rsos.royalsocietypublishing.org

algorithmic information theory/electrical engineering/mathematical modelling

Hilbert transform, Gabor analytic signal representation, instantaneous frequency, Fourier quadrature transform, Fourier quadrature analytic signal representations/Fourier–Singh analytic signal representations, Zero-phase filtering

**Author for correspondence:**
Pushpendra Singh
e-mail: spushp@gmail.com, pushpendrasingh@ iitkalumni.org

# Novel Fourier quadrature transforms and analytic signal representations for nonlinear and non-stationary time-series analysis

## Pushpendra Singh

School of Engineering and Applied Sciences, Bennett University, Greater Noida, India

PS, 0000-0001-5615-519X

The Hilbert transform (HT) and associated Gabor analytic signal (GAS) representation are well known and widely used mathematical formulations for modelling and analysis of signals in various applications. In this study, like the HT, to obtain quadrature component of a signal, we propose novel discrete Fourier cosine quadrature transforms (FCQTs) and discrete Fourier sine quadrature transforms (FSQTs), designated as Fourier quadrature transforms (FQTs). Using these FQTs, we propose 16 Fourier quadrature analytic signal (FQAS) representations with following properties: (1) real part of eight FQAS representations is the original signal, and imaginary part of each representation is FCQT of real part; (2) imaginary part of eight FQAS representations is the original signal, and real part of each representation is FSQT of imaginary part; (3) like the GAS, Fourier spectrum of all FQAS representations has only positive frequencies; however, unlike the GAS, real and imaginary parts of FQAS representations are not orthogonal. The Fourier decomposition method (FDM) is an adaptive data analysis approach to decompose a signal into a set Fourier intrinsic band functions. This study also proposes new formulations of the FDM using discrete cosine transform with GAS and FQAS representations, and demonstrates its efficacy for improved time-frequency-energy representation and analysis of many real-life nonlinear and non-stationary signals.

## 1. Introduction

The Fourier theory is the most important mathematical tool for analysis and modelling physical phenomena and engineering systems. It has been used to obtain solutions in almost all fields of science and engineering problems. It is the fundamentals of a

signal analysis, processing and interpretation of information. There are many variants of the Fourier methods such as continuous time Fourier series (FS) and Fourier transform (FT), discrete-time Fourier transform (DTFT), discrete-time Fourier series (DTFS), discrete Fourier transform (DFT), discrete cosine transforms (DCTs) and discrete sine transforms (DSTs). All these are orthogonal transforms which can be computed by the fast Fourier transform (FFT) algorithms.

The instantaneous frequency (IF) was introduced by the Carson & Fry [1] in 1937 with application to the frequency modulation (FM), and it was defined as derivative of phase of a complex FM signal. Gabor [2] in 1946 introduced a quadrature method based on the Fourier theory as a practical approach for obtaining the Hilbert Transform (HT) of a signal. Ville [3] in 1948 defined the IF of a real signal by using Gabor complex extension. Shekel [4] in 1953 pointed out ambiguity issue in the IF defined by Ville that there are an infinite number of pairs of instantaneous amplitude (IA) and IF for a complex extension of a given signal. Gabor analytic signal (GAS) representation, that has only positive frequencies in the Fourier spectrum which are identical to that of the real signal, is the fundamental principle of time-frequency analysis. In order to constrain the ambiguity issue, Vakman [5] in 1972 had shown that the GAS is the only physically justifiable complex extension for IA and IF estimation, and proposed the following three conditions to physical reality: (i) amplitude continuity, (ii) phase independence of scaling and homogeneity, and (iii) harmonic correspondence. Vakman also showed that the HT is the only operator that satisfies these conditions, thus, the unique complex extension can be obtained by the GAS representation. Therefore, almost universally, the HT has been used to construct the GAS representation and time-frequency analysis of a non-stationary signal. Several other authors contributed to representation and understanding of IF and Hilbert spectrum analysis, and have shown that there are problems and paradoxes related to the definition of IF [6–14]. In this study, using the DCTs and DSTs which are based on the Fourier theory, we propose the Fourier quadrature analytic signal (FQAS) or Fourier–Singh analytic signal (FSAS) representations. The proposed FSAS representations satisfy all the above three conditions of Vakman.

The DCT was proposed in the seminal paper [15] with application to image processing for pattern recognition and Wiener filtering. The modified DCT (MDCT), proposed in [16], is based on the DCT of overlapping data and uses the concept of time-domain aliasing cancellation [17]. Owing to energy compaction and decorrelation property of the DCT and MDCT, they are extensively used in many audio (e.g. MP3, WMA, AC-3, AAC, Vorbis, ATRAC), image (e.g. JPEG) and video (e.g. Motion JPEG, MPEG, Daala, digital video, Theora) compression, electrocardiogram (ECG) data analysis [18], reconstruction of financial time-series using DCT-based compressive sampling [19], and for numerical solution of partial differential equations by spectral methods. Depending upon the boundary conditions and symmetry about a data point, there are eight types of DCTs and eight types of DSTs.

Many real-life signals such as speech and animal sounds, mechanical vibrations, seismic wave, radar signals, biomedical ECG and electroencephalogram (EEG) signals are non-stationary and generated by nonlinear systems. These data can be characterized and modelled as superposition of amplitude-modulated–frequency-modulated (AM–FM) signals. Thus, signal decomposition, mode and source separation are important in many applications where received signal is the superposition of various non-stationary signals and noise, and the objective of study is to recover the original AM–FM constituents. There are many adaptive signal decomposition and analysis methods such as empirical mode decomposition (EMD) algorithms [10,20–26], Wigner distribution-based technique with reduced cross terms [27], variational mode decomposition (VMD) [28], synchrosqueezed wavelet transforms [29], eigenvalue decomposition approaches [30–32], Fourier-Bessel series expansion-based empirical wavelet transform [33], sparse time-frequency representation [34], improved eigenvalue decomposition and HT-based time-frequency distribution [35], tunable-Q wavelet transform and Wigner–Ville distribution-based TFR [36], empirical wavelet transform (EWT) [37], resonance-based signal decomposition (RSD) [38] and time-varying vibration decomposition (TVVD) [39]. These methods are developed based on the perception which has been for many decades in the literature that Fourier method is not suitable for nonlinear and non-stationary data analysis. However, the Fourier decomposition method (FDM) proposed in [14,40] is an adaptive, nonlinear and non-stationary data analysis method based on the Fourier theory and zero-phase filtering (ZPF) approach. The FDM can decompose real as well as complex signals (which can be multichannel or multivariate) into a set of desired number of Fourier intrinsic band functions (FIBFs) with desired cut-off frequencies [41]. The FDM has demonstrated its efficacy for representation and analysis of nonlinear and non-stationary data in many applications [14,40,42–44]. In this work, we consider the type-2 DCT [15], which is the most common variant of DCTs, to formulate the FDM. In principle, we can use any variant of DCTs or DSTs to formulate the FDM.

All these methods decompose the time-domain signal into a set of small number of band-limited components and map them into the time-frequency representation (TFR). The TFR provides localized

signal information in both time and frequency domain that reveal the complex structure of a signal consisting of several components. The IF is the basis of the TFR or time-frequency-energy (TFE) representation and analysis of a signal. The IF is a generalization of the definition of the traditional constant frequency, which is required for the analysis of non-stationary signals and nonlinear systems. It is an important parameter of a signal that can reveal the underlying process and provides explanations for physical phenomenon in many applications such as atmospheric and meteorological applications [10], image processing for pattern recognition and classification [45,46], mechanical systems analysis [47], acoustic, vibration and speech signal analysis [14], communications, radar, sonar, solar and seismic data analysis [14,48], medical and biomedical applications [49], time-frequency representation of cosmological gravity wave [50], patient-specific EEG seizure detection [51], epoch detection from speech signals [52], instantaneous fundamental frequency estimation from voiced speech [53], baseline wander and power line interference removal from ECG signals [54].

The main contributions [55] of this study are summarized as follows:

(1) Introduction of eight discrete Fourier cosine quadrature transforms (FCQTs) and eight discrete Fourier sine quadrature transforms (FSQTs) using eight DCTs and eight DSTs, respectively. These FCQTs and FSQTs are designated as the Fourier quadrature transforms (FQTs) and thus 16 FQTs are obtained.
(2) Introduction of the 16 FSAS representations, i.e. eight DCT-based analytic signal representations (DCT-ASRs) and eight DST-based ASRs (DST-ASRs), using 16 FQTs and corresponding DCTs/DSTs: (a) eight FSAS representations are obtained using the eight DCTs and corresponding FCQTs, where real part of DCT-ASRs is the original signal, imaginary part of each representation is the FCQT of the real part, (b) other eight FSAS representations are obtained using the eight DSTs and corresponding FSQTs, where imaginary part of DST–ASRs is the original signal, real part of each representation is FSQT of the imaginary part. In all the 16 FSAS representations, the Fourier spectrum has only positive frequencies; moreover, the real and imaginary parts are not orthogonal to each other.
(3) Introduction of the two continuous time FQTs, i.e. FCQT and FSQT, using the Fourier cosine transform (FCT) and Fourier sine transform (FST), respectively. Using these two FQTs corresponding FSAS representations are derived. The FQT and FSAS representation corresponding to the two-dimensional-DCT are also presented.
(4) The new formulations of the FDM are proposed using DCT and DST with GAS and FSAS representations. The FDM is computationally efficient due to FFT implementation, produces desired number of FIBFs with required cut-off frequencies. These attributes of the FDM are useful in many applications.

Thus, in this study, we present FQTs as effective alternatives to the HT, and FSAS representations as alternatives to the GAS representation for nonlinear and non-stationary time-series analysis. Generally, we acquire a continuous time (CT) signal and convert it to discrete-time (DT) signal for efficient compression, storage, transmission, representation and analysis. So, we consider and discuss only DT representations in detail, and discussion of its CT counterpart is limited, but can be easily obtained by analogy to the DT part. This study is organized as follows: a brief overview of the analytic signal representation and the FDM is presented in §2. FQTs, FSAS representations and new formulations of the FDM are presented in §3. Simulation results and discussions are presented in §4. Section 5 presents conclusion of the work.

## 2. A brief overview of the GAS representation, IF and FDM

The GAS representation [2] is a complex-valued function, $z[n]$, that has only positive frequency components in the Fourier spectrum, and it is defined as

$$z[n] = x[n] + j\hat{x}[n], \tag{2.1}$$

where the real part of GAS is the original signal and the imaginary part is the HT of the real part, and real and imaginary parts are orthogonal to each other (i.e. inner product $\langle x[n], \hat{x}[n] \rangle = 0$). The HT of a signal is defined as

$$\hat{x}[n] = H\{x[n]\} = x[n] * h[n] = \sum_{m=-\infty}^{\infty} x[m]h[n-m], \quad h[n] = \frac{1 - \cos(\pi n)}{\pi n}, \tag{2.2}$$

where * is the convolution operation, $H$ is the Hilbert operator and impulse response $h[n]$ is the Hilbert kernel. From (2.2), one can observe that the HT is an ideal operator which cannot be

rsos.royalsocietypublishing.org R. Soc. open sci. **5**: 181131

implemented in real applications, because, its impulse response is unstable (i.e. absolutely not summable as $\sum_{n=-\infty}^{\infty} |h[n]| = \infty$), non-causal and has infinite time support. Practically, the GAS representation $z[n]$, from a real signal $x[n]$ of length $N$, is obtained by the inverse discrete Fourier transform (IDFT) as [14,56]

$$z[n] = X[0] + \sum_{k=1}^{N/2-1} 2X[k]\exp\left(\frac{j2\pi kn}{N}\right) + X\left[\frac{N}{2}\right]\exp(j\pi n), \quad \text{if } N \text{ is even}$$

and

$$z[n] = X[0] + \sum_{k=1}^{(N-1)/2} 2X[k]\exp\left(\frac{j2\pi kn}{N}\right), \qquad \text{if } N \text{ is odd,} \tag{2.3}$$

where $X[k] = (1/N)\sum_{n=1}^{N-1} x[n]\exp(-j2\pi kn/N)$ is the DFT of a signal $x[n]$, $0 \le n, k \le N-1$, $\exp(j\pi n) = (-1)^n$, and $X[N/2]$ is the highest frequency component of the Fourier spectrum. This is the only practical approach, based on the Fourier theory and being used by Matlab as well, which is available in the literature to obtain the GAS representation that satisfies the following properties: **(P1)** only positive frequencies are present in the Fourier spectrum, **(P2)** real part is the original signal $x[n]$, **(P3)** imaginary part is the HT of real part (i.e. $\hat{x}[n] = H\{x[n]\}$), **(P4)** real and imaginary parts are orthogonal to each other (i.e. $\langle x[n], \hat{x}[n]\rangle = 0$) and **(P5)** real part is the HT of imaginary part with minus sign (i.e. $x[n] = -H\{\hat{x}[n]\} = -H^2\{x[n]\}$ or $x[n] = H^4\{x[n]\}$).

The GAS (2.1) can be written in polar representation as

$$z[n] = x[n] + j\hat{x}[n] = a[n]\exp(j\phi[n]), \tag{2.4a}$$

$$a[n] = \sqrt{x^2[n] + \hat{x}^2[n]}, \tag{2.4b}$$

$$\phi[n] = \arctan\left(\frac{\hat{x}[n]}{x[n]}\right) \tag{2.4c}$$

and

$$f[n] = \frac{\omega[n]}{2\pi} = \frac{\phi_d[n]}{2\pi}, \tag{2.4d}$$

where $a[n]$, $\phi[n]$ and $f[n]$ are the IA, instantaneous phase (IP) and the IF, respectively. The IF using differentiation of phase in DT, $\phi_d[n]$, can be approximated by Boashash [9] forward finite difference (FFD) or backward finite difference (BFD) or central finite difference (CFD) as

$$\phi_d[n] = (\phi[n+1] - \phi[n]), \quad \text{(FFD)} \tag{2.5a}$$

$$\phi_d[n] = (\phi[n] - \phi[n-1]), \quad \text{(BFD)} \tag{2.5b}$$

and

$$\phi_d[n] = \frac{(\phi[n+1] - \phi[n-1])}{2}, \quad \text{(CFD)}. \tag{2.5c}$$

The phase in (2.4c) is computed by the function, atan2($\hat{x}[n], x[n]$), which produces the result in the range $(-\pi, \pi)$ and also avoids the problems of division by zero. It is pertinent to notice that the IF defined by (2.4d) is valid only for monocomponent signals because the so-defined IF becomes negative in some time instants for multicomponent signals, which does not provide any physical meaning [8–10]. In order to eliminate this issue and obtain IF positive for all the time, by considering the phase unwrapping fact and *multivalued* nature of the inverse tangent function (i.e. $\tan(\phi[n]) = \tan(\phi[n] + kn\pi), \forall k, n \in \mathbb{Z}$), the IF $\omega[n]$ is defined as [41]

$$\omega[n] = \begin{cases} \phi_d[n], & \text{if } \phi_d[n] \ge 0, \\ \phi_d[n] + \pi, & \text{otherwise.} \end{cases} \tag{2.6}$$

This definition (2.6) makes the IF positive (i.e. $0 \le \omega[n] \le \pi$ in radians/sample which corresponds to $[0, F_s/2)$ in hertz) for all time ($n$) which is valid for all monocomponent as well as multicomponent signals.

The FDM is an adaptive signal decomposition approach which decomposes a signal, $x[n]$, into a set of small number of $M$ analytic Fourier intrinsic band functions (AFIBFs) such that

$$z[n] = a_0 + \sum_{i=1}^{M}(x_i[n] + j\hat{x}_i[n]) = a_0 + \sum_{i=1}^{M} a_i[n]\exp(j\phi_i[n]), \tag{2.7}$$

where $a_0 = X[0]$ is the average value of the signal, and $x_i[n] = a_i[n]\cos(\phi_i[n])$, $1 \le i \le M$ are amplitude-frequency modulated FIBFs which are complete, adaptive, local, orthogonal and uncorrelated by the virtue of construction [14].

rsos.royalsocietypublishing.org R. Soc. open sci. 5: 181131

In the next section, we propose a set of analytic signals using DCTs and DSTs which satisfy only the first two properties (P1) and (P2).

# 3. Fourier quadrature transforms, FSAS representations and new formulations of the FDM

The standard notations for the elements of the DCTs/DSTs transform matrices, $C_i/S_i$ for $i = 1, 2, \ldots, 8$, with their $nk$th element, denoted by $(C_i)_{nk}/(S_i)_{nk}$, are defined as [57]

$$
\left.
\begin{aligned}
&(\mathbf{C}_1)_{nk} = a\gamma_n\gamma_k \cos\left(\frac{nk\pi}{N-1}\right), &&(\mathbf{C}_2)_{nk} = b\sigma_k \cos\left[\left(n+\frac{1}{2}\right)\frac{k\pi}{N}\right], \\
&(\mathbf{C}_3)_{nk} = b\sigma_n \cos\left[\left(k+\frac{1}{2}\right)\frac{n\pi}{N}\right], &&(\mathbf{C}_4)_{nk} = b \cos\left[\left(n+\frac{1}{2}\right)\left(k+\frac{1}{2}\right)\frac{\pi}{N}\right], \\
&(\mathbf{C}_5)_{nk} = c\sigma_n\sigma_k \cos\left(\frac{nk2\pi}{2N-1}\right), &&(\mathbf{C}_6)_{nk} = c\varepsilon_n\sigma_k \cos\left[\left(n+\frac{1}{2}\right)\frac{k2\pi}{2N-1}\right], \\
&(\mathbf{C}_7)_{nk} = c\varepsilon_k\sigma_n \cos\left[\left(k+\frac{1}{2}\right)\frac{n2\pi}{2N-1}\right], &&(\mathbf{C}_8)_{nk} = d \cos\left[\left(n+\frac{1}{2}\right)\left(k+\frac{1}{2}\right)\frac{2\pi}{2N+1}\right], \\
&(\mathbf{S}_1)_{nk} = b \sin\left(\frac{nk\pi}{N}\right), &&(\mathbf{S}_2)_{nk} = b\varepsilon_k \sin\left[\left(n+\frac{1}{2}\right)\frac{(k+1)\pi}{N}\right], \\
&(\mathbf{S}_3)_{nk} = b\varepsilon_n \sin\left[\left(k+\frac{1}{2}\right)\frac{(n+1)\pi}{N}\right], &&(\mathbf{S}_4)_{nk} = b \sin\left[\left(n+\frac{1}{2}\right)\left(k+\frac{1}{2}\right)\frac{\pi}{N}\right], \\
&(\mathbf{S}_5)_{nk} = c \sin\left(\frac{nk2\pi}{2N-1}\right), &&(\mathbf{S}_6)_{nk} = c \sin\left[\left(n+\frac{1}{2}\right)\left(k+\frac{1}{2}\right)\frac{2\pi}{2N-1}\right], \\
&(\mathbf{S}_7)_{nk} = c \sin\left[\left(k+\frac{1}{2}\right)\frac{(n+1)2\pi}{2N-1}\right], &&(\mathbf{S}_8)_{nk} = c\varepsilon_n\varepsilon_k \sin\left[\left(n+\frac{1}{2}\right)\left(k+\frac{1}{2}\right)\frac{2\pi}{2N-1}\right],
\end{aligned}
\right\} \quad (3.1)
$$

where constant multiplication factors $a = \sqrt{2/(N-1)}$, $b = \sqrt{2/N}$, $c = 2/\sqrt{2N-1}$ and $d = 2/\sqrt{2N+1}$; normalization factors are unity except for $\gamma_n = \gamma_k = 1/\sqrt{2}$ for $n = k = 0$ or $N-1$, $\sigma_n = \sigma_k = 1/\sqrt{2}$ for $n = k = 0$, and $\varepsilon_n = \varepsilon_k = 1/\sqrt{2}$ for $n = k = N-1$; $0 \leq n, k \leq N-1$ for all the $N$th-order DCTs/DSTs except for the $(N-1)$th-order DST-1 and DST-5 where $1 \leq n, k \leq N-1$. As the DCTs and DSTs are unitary transform, their inverses are computed by transpose relation $C_i^{-1} = C_i^{\mathrm{T}}$ and $S_i^{-1} = S_i^{\mathrm{T}}$, respectively.

Using (3.1), we define the elements of transform matrices as follows:

$$
\left.
\begin{aligned}
&(\tilde{\mathbf{S}}_1)_{nk} = a\gamma_n\gamma_k \sin\left(\frac{nk\pi}{N-1}\right), &&(\tilde{\mathbf{S}}_2)_{nk} = b\sigma_k \sin\left[\left(n+\frac{1}{2}\right)\frac{k\pi}{N}\right], \\
&(\tilde{\mathbf{S}}_3)_{nk} = b\sigma_n \sin\left[\left(k+\frac{1}{2}\right)\frac{n\pi}{N}\right], &&(\tilde{\mathbf{S}}_4)_{nk} = b \sin\left[\left(n+\frac{1}{2}\right)\left(k+\frac{1}{2}\right)\frac{\pi}{N}\right], \\
&(\tilde{\mathbf{S}}_5)_{nk} = c\sigma_n\sigma_k \sin\left(\frac{nk2\pi}{2N-1}\right), &&(\tilde{\mathbf{S}}_6)_{nk} = c\varepsilon_n\sigma_k \sin\left[\left(n+\frac{1}{2}\right)\frac{k2\pi}{2N-1}\right], \\
&(\tilde{\mathbf{S}}_7)_{nk} = c\varepsilon_k\sigma_n \sin\left[\left(k+\frac{1}{2}\right)\frac{n2\pi}{2N-1}\right], &&(\tilde{\mathbf{S}}_8)_{nk} = d \sin\left[\left(n+\frac{1}{2}\right)\left(k+\frac{1}{2}\right)\frac{2\pi}{2N+1}\right], \\
&(\tilde{\mathbf{C}}_1)_{nk} = b \cos\left(\frac{nk\pi}{N}\right), &&(\tilde{\mathbf{C}}_2)_{nk} = b\varepsilon_k \cos\left[\left(n+\frac{1}{2}\right)\frac{(k+1)\pi}{N}\right], \\
&(\tilde{\mathbf{C}}_3)_{nk} = b\varepsilon_n \cos\left[\left(k+\frac{1}{2}\right)\frac{(n+1)\pi}{N}\right], &&(\tilde{\mathbf{C}}_4)_{nk} = b \cos\left[\left(n+\frac{1}{2}\right)\left(k+\frac{1}{2}\right)\frac{\pi}{N}\right], \\
&(\tilde{\mathbf{C}}_5)_{nk} = c \cos\left(\frac{nk2\pi}{2N-1}\right), &&(\tilde{\mathbf{C}}_6)_{nk} = c \cos\left[\left(n+\frac{1}{2}\right)\left(k+\frac{1}{2}\right)\frac{2\pi}{2N-1}\right], \\
&(\tilde{\mathbf{C}}_7)_{nk} = c \cos\left[\left(k+\frac{1}{2}\right)\frac{(n+1)2\pi}{2N-1}\right], &&(\tilde{\mathbf{C}}_8)_{nk} = c\varepsilon_n\varepsilon_k \cos\left[\left(n+\frac{1}{2}\right)\left(k+\frac{1}{2}\right)\frac{2\pi}{2N-1}\right],
\end{aligned}
\right\} \quad (3.2)
$$

where matrices $\tilde{\mathbf{S}}_1$, $\tilde{\mathbf{S}}_2$, $\tilde{\mathbf{S}}_3$, $\tilde{\mathbf{S}}_5$, $\tilde{\mathbf{S}}_7$, $\tilde{\mathbf{C}}_1$ and $\tilde{\mathbf{C}}_5$ are of $(N-1)$th-order matrices, and rest are of $N$th-order.

Using (3.1) and (3.2), we hereby define 16 FQTs (i.e. eight FCQTs, $\tilde{\mathbf{x}}_{ci}$ and eight FSQTs, $\tilde{\mathbf{x}}_{si}$) and corresponding 16 FSAS representations (i.e. eight DCT-ASRs $\tilde{\mathbf{z}}_{ci}$ and eight DST-ASRs $\tilde{\mathbf{z}}_{si}$ for $i = 1, 2, \ldots, 8$) as follows:

$$
\left.
\begin{aligned}
\mathbf{X}_{ci} &= \mathbf{C}_i \mathbf{x}; & \mathbf{x} &= \mathbf{C}_i^{\mathrm{T}} \mathbf{X}_{ci}; & \text{(DCTs and IDCTs)} \\
\tilde{\mathbf{x}}_{ci} &= \tilde{\mathbf{S}}_i^{\mathrm{T}} \mathbf{X}_{ci} = \tilde{\mathbf{S}}_i^{\mathrm{T}} \mathbf{C}_i \mathbf{x}; & \tilde{\mathbf{z}}_{ci} &= \mathbf{x} + j\tilde{\mathbf{x}}_{ci}; & \text{(FCQTs and DCT-ASRs)} \\
\mathbf{X}_{si} &= \mathbf{S}_i \mathbf{x}; & \mathbf{x} &= \mathbf{S}_i^{\mathrm{T}} \mathbf{X}_{si}; & \text{(DSTs and IDSTs)} \\
\tilde{\mathbf{x}}_{si} &= \tilde{\mathbf{C}}_i^{\mathrm{T}} \mathbf{X}_{si} = \tilde{\mathbf{C}}_i^{\mathrm{T}} \mathbf{S}_i \mathbf{x}; & \tilde{\mathbf{z}}_{si} &= \tilde{\mathbf{x}}_{si} + j\mathbf{x}; & \text{(FSQTs and DST-ASRs),}
\end{aligned}
\right\}
\tag{3.3}
$$

and

where (column vectors) data $\mathbf{x} = [x[0]\ x[1] \ldots x[N-1]]^{\mathrm{T}}$; $\mathbf{X}_{ci} = [X_{ci}[0]\ X_{ci}[1] \ldots X_{ci}[N-1]]^{\mathrm{T}}$ and $\mathbf{X}_{si} = [X_{si}[0]\ X_{si}[1] \ldots X_{si}[N-1]]^{\mathrm{T}}$ are the DCT and DST of $i$th type, respectively. Thus, we have defined linear transformations of $\mathbf{x}$ into $\tilde{\mathbf{x}}_{ci}$, and $\mathbf{x}$ into $\tilde{\mathbf{x}}_{si}$ with orthogonal transformation matrices $\tilde{\mathbf{S}}_i^{\mathrm{T}} \mathbf{C}_i$ and $\tilde{\mathbf{C}}_i^{\mathrm{T}} \mathbf{S}_i$, respectively. These transformation matrices are orthogonal due to the properties of an orthogonal matrix (i.e. if $Q$ is an orthogonal matrix, then so is $Q^{\mathrm{T}}$ and $Q^{\mathrm{T}} = Q^{-1}$; if $Q_1$ and $Q_2$ are orthogonal matrices, then so is $Q_1 Q_2$). The continuous time FCQT, FSQT and corresponding FSAS representations are defined in appendix A. The proposed two-dimensional FSAS representations are defined in appendix B.

Now, we consider the complete process of obtaining FQT, corresponding FSAS and FDM using DCT-2 as follows. The DCT-2 of a sequence, $x[n]$ of length $N$, is defined as [15]

$$
X_{c2}[k] = \sqrt{\frac{2}{N}} \sigma_k \sum_{n=0}^{N-1} x[n] \cos\left(\frac{\pi k(2n+1)}{2N}\right), \quad 0 \le k \le N-1,
\tag{3.4}
$$

and inverse DCT (IDCT) is obtained by

$$
x[n] = \sqrt{\frac{2}{N}} \sum_{k=0}^{N-1} \sigma_k X_{c2}[k] \cos\left(\frac{\pi k(2n+1)}{2N}\right), \quad 0 \le n \le N-1,
\tag{3.5}
$$

where normalization factors $\sigma_k = 1/\sqrt{2}$ for $k = 0$, and $\sigma_k = 1$ for $k \ne 0$. If consecutive samples of sequence $x[n]$ are correlated, then DCT concentrates energy in a few $X_{c2}[k]$ and decorrelates them. The DCT basis sequences, $\cos(\pi k(2n+1)/2N)$, which are a class of discrete Chebyshev polynomials [15], form an orthogonal set as inner product $\langle \cos(\pi k(2n+1)/2N), \cos(\pi m(2n+1)/2N) \rangle = 0$ for $k \ne m$, and

$$
\sum_{n=0}^{N-1} \cos\left(\frac{\pi k(2n+1)}{2N}\right) \cos\left(\frac{\pi m(2n+1)}{2N}\right) = \begin{cases} N, & k = m = 0, \\ \frac{N}{2}, & k = m \ne 0, \\ 0, & k \ne m. \end{cases}
\tag{3.6}
$$

We hereby formally define the discrete FCQT, $\tilde{x}_{c2}[n]$, of a signal $x[n]$ as

$$
\tilde{x}_{c2}[n] = \sqrt{\frac{2}{N}} \sum_{k=0}^{N-1} X_{c2}[k] \sin\left(\frac{\pi k(2n+1)}{2N}\right), \quad 0 \le n \le N-1,
\tag{3.7}
$$

where $X_{c2}[k]$ is the DCT-2 of a signal $x[n]$. Since, for frequency $k = 0$, in (3.7), basis vector $\sin(\pi k(2n+1)/2N)$ is zero, we can write FCQT as $\tilde{x}_{c2}[n] = \sqrt{2/N} \sum_{k=1}^{N-1} X_{c2}[k] \sin(\pi k(2n+1)/2N)$ and obtain $\tilde{X}_{c2}[k]$ from (3.7) as

$$
\tilde{X}_{c2}[k] = \sqrt{\frac{2}{N}} \sum_{n=0}^{N-1} \tilde{x}_{c2}[n] \sin\left(\frac{\pi k(2n+1)}{2N}\right), \quad 0 \le k \le N-1,
\tag{3.8}
$$

where

$$
\tilde{X}_{c2}[k] = \begin{cases} 0, & k = 0, \\ X_{c2}[k], & 1 \le k \le N-1. \end{cases}
\tag{3.9}
$$

One can observe that $X_{c2}[k]$ defined in (3.4) and $\tilde{X}_{c2}[k]$ (3.8) are exactly the same for zero-mean signal (i.e. $X_{c2}[0] = 0$), otherwise, they are different only for $k = 0$. The IDCT of $\tilde{X}_{c2}[k]$ is original signal less DC component.

rsos.royalsocietypublishing.org   R. Soc. open sci. **5**: 181131

The proposed FCQT (3.7) uses the basis sequences, $\sin(\pi k(2n+1)/2N)$, whose inner product is defined as

$$\sum_{n=0}^{N-1} \sin\left(\frac{\pi k(2n+1)}{2N}\right) \sin\left(\frac{\pi m(2n+1)}{2N}\right) = \begin{cases} 0, & k=m=0, \\ N/2, & k=m \neq 0, \\ 0, & k \neq m, \end{cases} \tag{3.10}$$

and, therefore, the set of these basis vectors form an orthogonal set for $1 \leq k \leq N-1$. From (3.4) and (3.7), one can easily observe that the FCQT of a constant signal, like the HT, is zero.

The vector space theoretic approach and explanation of this proposed transformation is as follows. Let $V$ and $W$ be vector spaces, a function $T:V \rightarrow W$ is called a linear transformation if for any vectors $v_1, v_2 \in V$ and scalar $c$, (i) $T(v_1 + v_2) = T(v_1) + T(v_2)$ and (ii) $T(cv_1) = cT(v_1)$. Here in this study, $V$ is a vector space spanned by the set of cosine basis vectors, $\{\cos(\pi k(2n+1)/2N)\}$ for $0 \leq k \leq N-1$, with dimension $N$, and $W$ is a vector space spanned by the set of sine basis vectors, $\{\sin(\pi k(2n+1)/2N)\}$ for $0 \leq k \leq N-1$, with dimension $(N-1)$. Therefore, these two vector spaces are homomorphic, transformation is linear and non-invertible as a constant vector is mapped to the zero vector. However, for a zero-mean vector $x[n]$ (i.e. $X_{c2}[0] = 0$), these two vector spaces are isomorphic, transformation is linear and invertible, and a set of orthogonal cosine basis vectors are mapped to the set of orthogonal sine basis vectors, i.e. $\{\cos(\pi k(2n+1)/2N)\} \mapsto \{\sin(\pi k(2n+1)/2N)\}$ or $T(\cos(\pi k(2n+1)/2N)) = (\sin(\pi k(2n+1)/2N))$ for $1 \leq k \leq N-1$. Thus, the linear transformation of $x[n]$ into $\tilde{x}_{c2}[n]$, defined in (3.3) and (3.7) with transformation matrix $\tilde{\mathbf{S}}_2^{\mathrm{T}}\mathbf{C}_2$, is (a) non-invertible if the mean of signal $x[n]$ is non-zero and (b) invertible if the mean of signal $x[n]$ is zero. In practice, we can always remove the mean from $x[n]$ to make it a zero-mean vector that leads the proposed transformation to be isomorphic.

We proposed and derived 16 different types of FSAS representations in (3.3) using eight types of DCTs and eight types of DSTs. However, in this study, we especially consider DCT-2 in detail to obtain the FSAS as

$$\tilde{z}_{c2}[n] = \sqrt{\frac{2}{N}} \sum_{k=0}^{N-1} \sigma_k X_{c2}[k] \exp\left(j \frac{\pi k(2n+1)}{2N}\right) = x[n] + j\tilde{x}_{c2}[n], \quad 0 \leq n \leq N-1, \tag{3.11}$$

where $j = \sqrt{-1}$, real part of $\tilde{z}_{c2}[n]$ is the original signal, and imaginary part of $\tilde{z}_{c2}[n]$ is the FQT of real part. It is worthwhile to note that the signal $x[n]$ and its FQT $\tilde{x}_{c2}[n]$ are not orthogonal, i.e. $\langle x[n], \tilde{x}_{c2}[n] \rangle \neq 0$. To prove this, we consider the inner product of basis sequences, $\cos(\pi k(2n+1)/2N)$ and $\sin(\pi m(2n+1)/2N)$, and using trigonometric manipulation $[2\cos(\alpha)\sin(\beta) = \sin(\alpha+\beta) - \sin(\alpha-\beta)$, $\sin(2\alpha) = 2\sin(\alpha)\cos(\alpha)$, $2\sin^2(\alpha) = 1 - \cos(2\alpha)]$ show that it is not zero for some $k \neq m$, i.e.

$$\sum_{n=0}^{N-1} \cos\left(\frac{\pi k(2n+1)}{2N}\right) \sin\left(\frac{\pi m(2n+1)}{2N}\right)$$

$$= \begin{cases} 0, & k=m=0, \\ 0, & k=m \neq 0, \\ 0, & k \neq m, \text{ and } m \pm k = 2l, \\ \Sigma_{m,k} = \Sigma_{m+k} + \Sigma_{m-k}, & k \neq m, \text{ and } m \pm k = (2p+1), \end{cases} \tag{3.12}$$

where $\Sigma_{m\pm k} = \frac{1}{2}\sum_{n=0}^{N-1} \sin(\pi(2n+1)(m \pm k)/2N)$, $0 \leq l \leq (N-1)$ and $0 \leq p \leq (N-2)$. We obtain sum of exponential series as $E_{m\pm k} = \left[\sum_{n=0}^{N-1} \exp(j\pi(2n+1)(m \pm k)/2N)\right] = [\exp(j\pi(m \pm k)/2)$ $(\sin(\pi(m \pm k)/2)/\sin(\pi(m \pm k)/2N))]$, which implies $E_{m \pm k} = [\sin(\pi(m \pm k))2\sin(\pi(m \pm k)/2N) + j([1 - \cos(\pi(m \pm k))]/2\sin(\pi(m \pm k)/2N))]$. Thus, real part $\mathrm{Re}\{E_{m \pm k}\} = 0$ if $m \neq k$; imaginary part $\mathrm{Im}\{E_{m \pm k}\} = 0$ if $m \pm k = 2l$, and $\mathrm{Im}\{E_{m \pm k}\} = 1/\sin(\pi(m \pm k)/2N)$ if $m \pm k = 2p + 1$, $\Sigma_{m\pm k} = \frac{1}{2}\mathrm{Im}\{E_{m\pm k}\}$, and $|\Sigma_{m,k}| \rightarrow \infty$ when $N \rightarrow \infty$ with $m \pm k = 2p + 1$.

Now, we propose the DCT-based FDM (i.e. DCT–FDM), and devise two new approaches to decompose a signal into a small set of AM–FM signals (i.e. FIBFs) and corresponding analytic representations. In the first approach, we obtain the decomposition of signal $x[n]$ (excluding the DC component) into a set of FIBFs, $\{x_i[n]\}_{i=1}^{M}$, and corresponding AFIBFs, $\{\tilde{z}_{c2i}[n]\}_{i=1}^{M}$, using the FSAS representation (3.11) as

$$\tilde{z}_{c2}[n] = \sqrt{\frac{2}{N}} \sum_{k=1}^{N-1} X_{c2}[k] \exp\left(j \frac{\pi k(2n+1)}{2N}\right) = \sum_{i=1}^{M} \tilde{z}_{c2i}[n], \quad 0 \leq n \leq N-1, \tag{3.13}$$

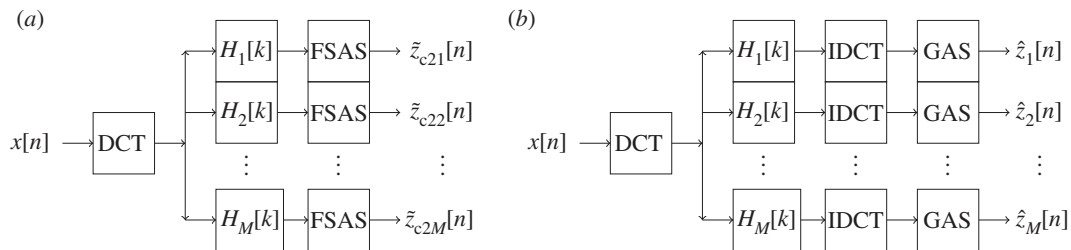

**Figure 1.** Block diagrams of the FDM using DCT-based zero-phase filter-bank to decompose a signal $x[n]$ into the set of orthogonal desired frequency bands ($a$) $\{\tilde{z}_{c21}[n], \tilde{z}_{c22}[n], \ldots, \tilde{z}_{c2M}[n]\}$ using (3.13) and (3.14); ($b$) $\{\hat{z}_1[n], \hat{z}_2[n], \ldots, \hat{z}_M[n]\}$ using (3.15) and (3.16).

where $\tilde{z}_{c21}[n] = \sqrt{2/N} \sum_{k=1}^{N_1} X_{c2}[k] \exp(j(\pi k(2n+1)/2N))$, $\tilde{z}_{c22}[n] = \sqrt{2/N} \sum_{k=N_1+1}^{N_2} X_{c2}[k] \exp(j(\pi k(2n+1)/2N)), \ldots, \tilde{z}_{c2M}[n] = \sqrt{2/N} \sum_{k=N_{M-1}+1}^{N_M} X_{c2}[k] \exp(j(\pi k(2n+1)/2N))$, which can be written as

$$\tilde{z}_{c2i}[n] = \sqrt{\frac{2}{N}} \sum_{k=N_{i-1}+1}^{N_i} X_{c2}[k] \exp\left(j\frac{\pi k(2n+1)}{2N}\right) = x_i[n] + j\tilde{x}_{c2i}[n], \quad 1 \le i \le M, \tag{3.14}$$

where $N_0 = 0$ and $N_M = N - 1$.

In the second approach, using the DCT–FDM, we decompose the signal $x[n]$ into the same set of FIBFs, $\{x_i[n]\}_{i=1}^M$, and apply the HT to obtain corresponding set of AFIBFs, $\{\hat{z}_i[n]\}_{i=1}^M$, using the GAS representation as

$$\hat{z}[n] = \sum_{i=1}^M \hat{z}_i[n], \quad \hat{z}_i[n] = x_i[n] + j\hat{x}_i[n], \quad 1 \le i \le M, \tag{3.15}$$

where $\hat{x}_i[n]$ is the HT of $x[n]$, $\langle x_i[n], \hat{x}_i[n]\rangle = 0$, and

$$x_i[n] = \sqrt{\frac{2}{N}} \sum_{k=N_{i-1}+1}^{N_i} X_{c2}[k] \cos\left(\frac{\pi k(2n+1)}{2N}\right). \tag{3.16}$$

It is pertinent to note that in the FDM (3.15) all five properties (P1)–(P5) of the GAS are satisfied; however, in the FDM (3.14), which uses FSAS, only the first two properties (P1) and (P2) of the GAS are being satisfied.

The block diagrams of the FDM, using DCT-based zero-phase filter-bank to decompose a signal into a set of desired frequency bands, are shown in figure 1, where, for each $i \in [1, M]$,

$$H_i[k] = \begin{cases} 1, & (N_{i-1} + 1) \le k \le N_i, \\ 0, & \text{otherwise.} \end{cases} \tag{3.17}$$

We have used ZPF where frequency response of filter is one in the desired band and zero otherwise. Moreover, one can use any other kind of ZPF (i.e. $H_i[k] \in \mathbb{R}_{\ge 0}, \forall i, k$) such as the Gaussian filter to decompose a signal. The FDM advocates to use ZPF because it preserves salient features such as minima and maxima in the filtered waveform exactly at the position where those features occur in the original unfiltered waveform. Based on the requirement and type of application, we can devise methods to select ranges of frequency parameter $k$ in (3.17), e.g. one can use three approaches to divide complete frequency band of a signal under analysis into small number of equal, dyadic and equal-energy frequency bands.

To obtain TFE representation of a signal using (3.13) and (3.15), we write FSAS and GAS in polar representation, for $1 \le i \le M$, as

$$\tilde{z}_{c2i}[n] = \tilde{a}_i[n] e^{j\tilde{\phi}_i[n]}, \quad \tilde{\phi}_i[n] = \tan^{-1}\left(\frac{\tilde{x}_{c2i}[n]}{x_i[n]}\right), \quad \tilde{\omega}_i[n] = \tilde{\phi}_{id}[n] \tag{3.18a}$$

and

$$\hat{z}_i[n] = \hat{a}_i[n] e^{j\hat{\phi}_i[n]}, \quad \hat{\phi}_i[n] = \tan^{-1}\left(\frac{\hat{x}_i[n]}{x_i[n]}\right), \quad \hat{\omega}_i[n] = \hat{\phi}_{id}[n], \tag{3.18b}$$

where IA ($\tilde{a}_i[n]$, $\hat{a}_i[n]$) and IF ($\tilde{\omega}_i[n]$, $\hat{\omega}_i[n]$) are computed by (2.4$b$) and (2.4$d$), respectively. Finally, using the proposed FSAS and the GAS representations, the three-dimensional TFE distributions are obtained by plotting $\{n, \tilde{f}_i[n], \tilde{a}_i^2[n]\}$ and $\{n, \hat{f}_i[n], \hat{a}_i^2[n]\}$, respectively.

---

**Algorithm 1.** Proposed FDM corresponding to figure 1, for $i = 1, \ldots, M$.

1. Obtain and decide the desired number of AFIBFs with their cut-off frequencies (e.g. 10-AFIBFs and dyadic frequency bands ($[0, \frac{F_s}{2^{10}}], \ldots, [\frac{F_s}{2^2}, \frac{F_s}{2}]$, where $F_s$ is a sampling frequency), then using (3.17) we obtain filter $H_i[k]$, where $N_i = \text{round}(\frac{N}{2^{10-i}})$ for $1 \le i \le 10$, in this example).
2. Obtain DCT of $x[n]$ using (3.4), i.e. $X_{c2}[k] = \text{DCT}(x[n])$.
3. Set $X_i[k] = X_{c2}[k]H_i[k]$, and using IDCT (3.5) obtain $x_i[n] = \text{IDCT}(X_i[k])$.
4. From $X_i[k]$ obtain FSAS $\tilde{z}_{c2i}[n]$ using (3.11); and GAS $\hat{z}_i[n] = x_i[n] + j\hat{x}_i[n]$ using (2.3).
5. Repeat steps 3–4 for $i = 1, \ldots, M$.

---

Algorithms to implement FDM using the DFT and finite impulse response (FIR) filtering are presented in [14,41]. We summarize the steps to implement the proposed FDM, using DCT to decompose a signal $x[n]$ into a set of $M$-AFIBFs, in algorithm 1, and complete Matlab implementation of the FDM using DFT, DCT and FIR filtering is made publicly available to download at the Dryad Digital Repository at https://doi.org/10.5061/dryad.jc21t36 [58] and at https://www.rese archgate.net/publication/326294577_MATLABCodeOfFDM_DCT_DFT_FIR_FSASJuly2018. From figure 1 as well as algorithm 1, we observe that FDM implementation requires one DCT/DFT and $M$ number of IDCT/IDFT, so computational complexity is $(M + 1)$ times of FFT algorithm that has complexity of order $\mathcal{O}(N \log_2 N)$.

# 4. Results and discussions

In this section, to demonstrate the efficacy of the proposed methods—DCT- and FSAS-based FDM (DCT–FSAS–FDM), DCT- and GAS-based FDM (DCT–GAS–FDM)—we consider many simulated as well as real-life data, and compare the obtained results with other popular methods such as EMD-, CWT-, DFT- and GAS-based FDM (DFT–GAS–FDM), FIR- and GAS-based FDM (FIR–GAS–FDM). We primarily consider those signals which have been widely used in literature for performance evaluation and results comparison among proposed and other existing methods.

## 4.1. A unit sample sequence analysis

First, we consider an analysis of unit sample (or unit impulse or delta) sequence using the proposed FSAS and compare the results with GAS representation. A unit sample function is defined as $\delta[n - n_0] = 1$ for $n = n_0$, and $\delta[n - n_0] = 0$ for $n \ne n_0$. Using the inverse Fourier transform, analytic representation, $z[n] = (1/\pi) \int_0^\pi X(\omega) \exp(j\omega n) \, d\omega$, of a signal $x[n] = \delta[n - n_0] \Leftrightarrow X(\omega) = \exp(-j\omega n_0)$ is obtained as [14] $z[n] = (\sin(\pi(n - n_0)) + j[1 - \cos(\pi(n - n_0))])/\pi(n - n_0) = a[n] \exp(j\phi[n])$, where real part of $z[n]$ is $\delta[n - n_0] = \sin(\pi(n - n_0))/\pi(n - n_0), a[n] = \sin((\pi/2)(n - n_0))/((\pi/2)(n - n_0)))$ (IA), $\phi[n] = (\pi/2)(n - n_0)$ (IP), and therefore $\omega[n] = \pi/2$ (IF) which corresponds to half of the Nyquist frequency, that is, $F_s/4$ Hz. Theoretically, this representation demonstrates that most of the energy of signal $\delta[n - n_0]$ is concentrated at time $t = 4.99$ s ($n_0 = 499$) and frequency $f = 25$ Hz, where sampling frequency, $F_s = 100$ Hz, and length of signal, $N = 1000$, are considered. The plots of IA, IF and TFE using the GAS and FSAS representations are shown in figure 2$a$,$b$, respectively. In the GAS representation, figure 2$a$, IF is varying between 0 and 50 Hz at both ends of signal and converges to theoretical value only at the position of delta function. On the other hand, the FSAS representation yields better results as it produces correct value of IF, figure 2$b$, for all the time.

## 4.2. Chirp signal analysis

In this example, we consider an analysis of a non-stationary chirp signal using the proposed FSAS and compare the results with GAS approach, with sampling frequency $F_s = 1000$ Hz, time duration $t \in [0, 1)$ s and frequency $f \in [5, 100]$ Hz. Figure 3$a$ shows chirp signal (5–100 Hz) (i), proposed FCQT (ii) obtained using (3.7), which is also imaginary part of the proposed FSAS (3.11), and HT (iii) that is imaginary part of the GAS representation (2.3), which has unnecessary distortions at both ends of the signal. The TFE distributions obtained (i) using the FSAS representation is shown in figure 3$b$ and (ii) using the GAS representation is shown in figure 3$c$, which has lot of energy spreading over wide range of time–frequency plane, at both ends of the signal under analysis.

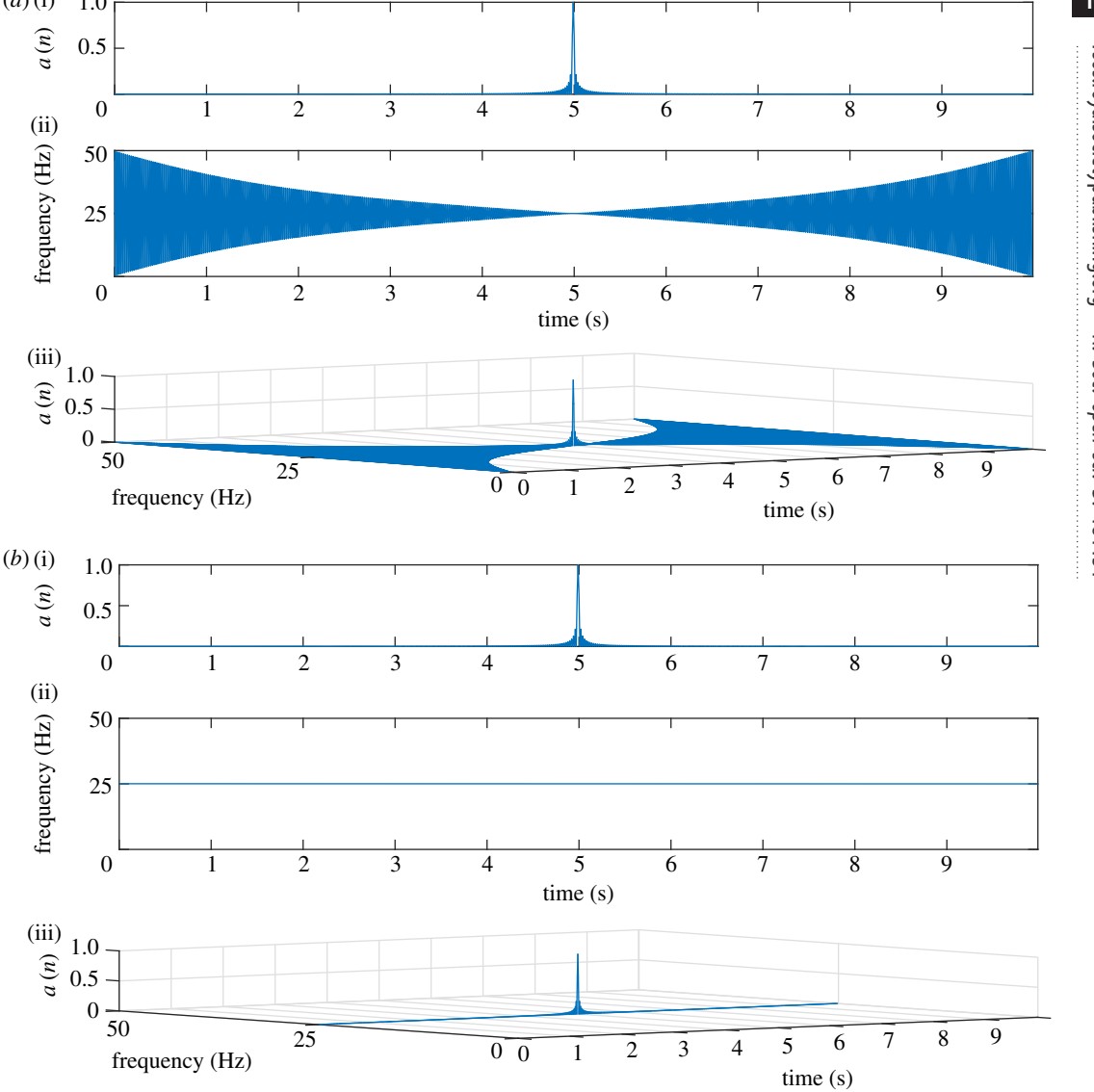

**Figure 2.** Analysis of unit sample sequence: time-amplitude (i), time-frequency (ii) and time-frequency-amplitude (iii) subplots (*a*) and (*b*) obtained using the GAS and FSAS representations, respectively.

rsos.royalsocietypublishing.org    R. Soc. open sci. **5**: 181131

These examples, §§4.1 and 4.2, clearly demonstrate the advantages of the proposed FSAS over the GAS representation for IF estimation and TFE analysis of a signal.

## 4.3. Speech signal analysis

Estimation of the instantaneous fundamental frequency $F_0$ is the most important problem in speech processing, because it arises in numerous applications, such as language identification [58], speaker recognition [59], emotion analysis [60], speech compression and voice conversion [61,62]. Figure 4*a* shows a segment of voiced speech signal (i) from the CMU Arctic database [63] sampled at $F_s = 32$ 000 Hz, corresponding electroglottograph (EGG) signal (ii) and differenced EGG (iii) signal. Figure 4*b*(i) shows magnitude spectrum of speech signal where $F_0$ is around 132 Hz, and energy of speech signal is concentrated on harmonics of $F_0$, and the magnitude spectrum of differenced EGG (DEGG) signal is shown in plot (ii) that also indicate $F_0$ around 132 Hz. Figure 4*c* shows a decomposition of speech segment into a set of 10 FIBFs (FIBF1–FIBF10), a high-frequency component (FIBF11) and a low-frequency component (LFC) using the FDM, where FIBF1 captures $F_0$, and harmonic components of $F_0$ are encapsulated in FIBF2–FIBF10. Figure 4*d* presents TFE representation obtained from the FDM where $F_0$ and its harmonics are clearly separated in distinct frequency bands.

Jain & Pachori [53] studied event-based method for $F_0$ estimation from voiced speech based on eigenvalue decomposition of Hankel matrix, and method uses iterative approach to estimate $F_0$, which

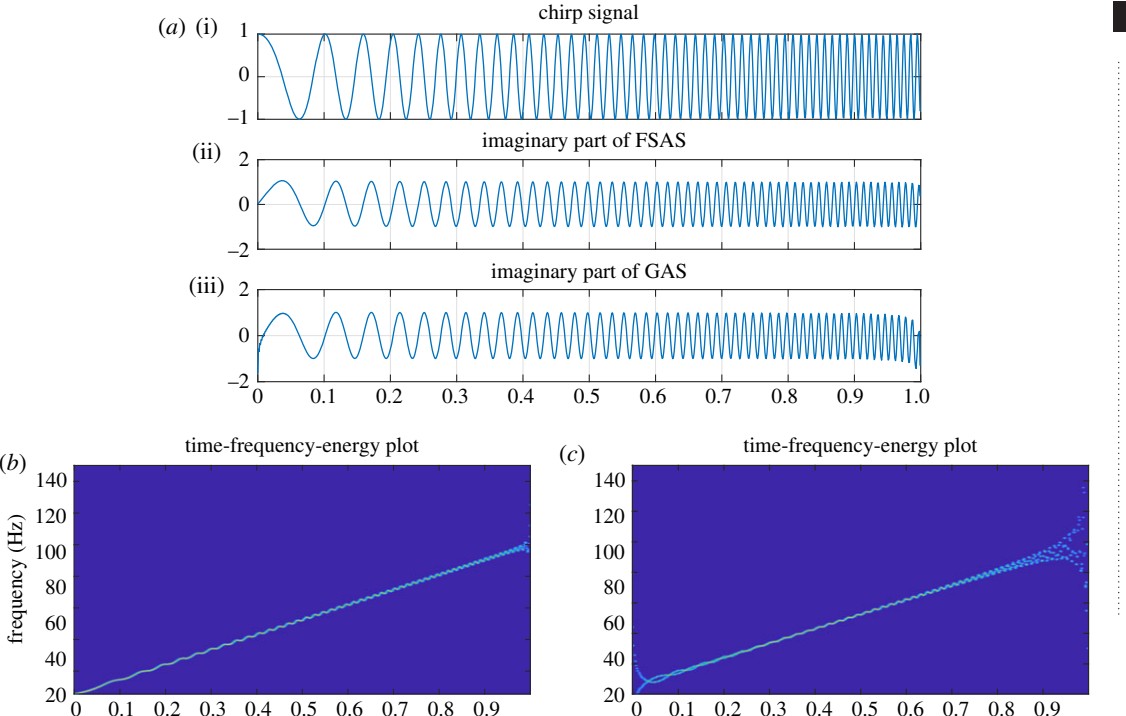

**Figure 3.** Analysis of a non-stationary signal: (*a*) a chirp signal (5 – 100 Hz) (i), FCQT-imaginary part of the proposed FSAS (ii), HT-imaginary part of GAS (iii); obtained TFE representations using the FSAS (*b*) and GAS (*c*).

is computationally complex; moreover, it is difficult for method to estimate all harmonics of $F_0$ accurately. On the other hand, the FDM-based approach presented in this study is able and efficient to implement, achieve and follow the proposed model (2.7) precisely.

## 4.4. Noise removal from ECG signal

ECG signals, records of the electrical activity of the heart, are used to examine the activity of human heart. There are various problems that may arise while recording an ECG signal. It may be distorted due to the presence of various noises such as channel noise, baseline wander (BLW) noise (of generally below 0.5 Hz), power-line interference (PLI) of 50 Hz (or 60 Hz) and physiological artefacts. Owing to these noises, it becomes difficult to diagnose diseases, and thus appropriate treatment may be impacted [64]. The BLW and PLI removal from an ECG signal (obtained from MIT-Arrhythmia database, $Fs = 360$ Hz) using the proposed method is shown in figure 5: (*a*) a segment of clean ECG signal (i), ECG signal heavily corrupted (i.e. signal-to-noise ratio (SNR) is low, in the range $\approx -18.4$ dB) with BLW and PLI (ii) noises, (*b*) ECG signal after noise removal (i), separated BLW (ii) and PLI (iii). Thus, proposed FDM can be used to remove BLW and PLI noises, and recover ECG signal even in scenarios where SNR is rather poor. To present a numerical comparison, we consider input $SNR_i$ and output $SNR_o$, which are defined as

$$SNR_i = 10 \log \left( \frac{\sum_{n=0}^{N-1} x^2[n]}{\sum_{n=0}^{N-1} w^2[n]} \right), \quad SNR_o = 10 \log \left( \frac{\sum_{n=0}^{N-1} x^2[n]}{\sum_{n=0}^{N-1} (x[n] - \tilde{x}[n])^2} \right), \tag{4.1}$$

respectively, where $x[n]$ is the original ECG signal, $w[n]$ is the sum of BLW and PLI components, and $\tilde{x}[n]$ is the estimated ECG signal, i.e. after removing BLW and PLI noises, using the EMD algorithm and FDM. A comparison of the EMD and FDM approaches using the input and output SNR criterion is shown in figure 5*c*, which clearly demonstrates that the output SNR is best for FDM with DFT/DCT approach, followed by FDM with zero-phase FIR filtering approach, and in all cases FDM outperforms the widely used EMD algorithm. Moreover, authors in [54] studied the BLW and PLI removal from ECG signals and obtained best-case results which are presented (from table 6 of [54]) here as follows:

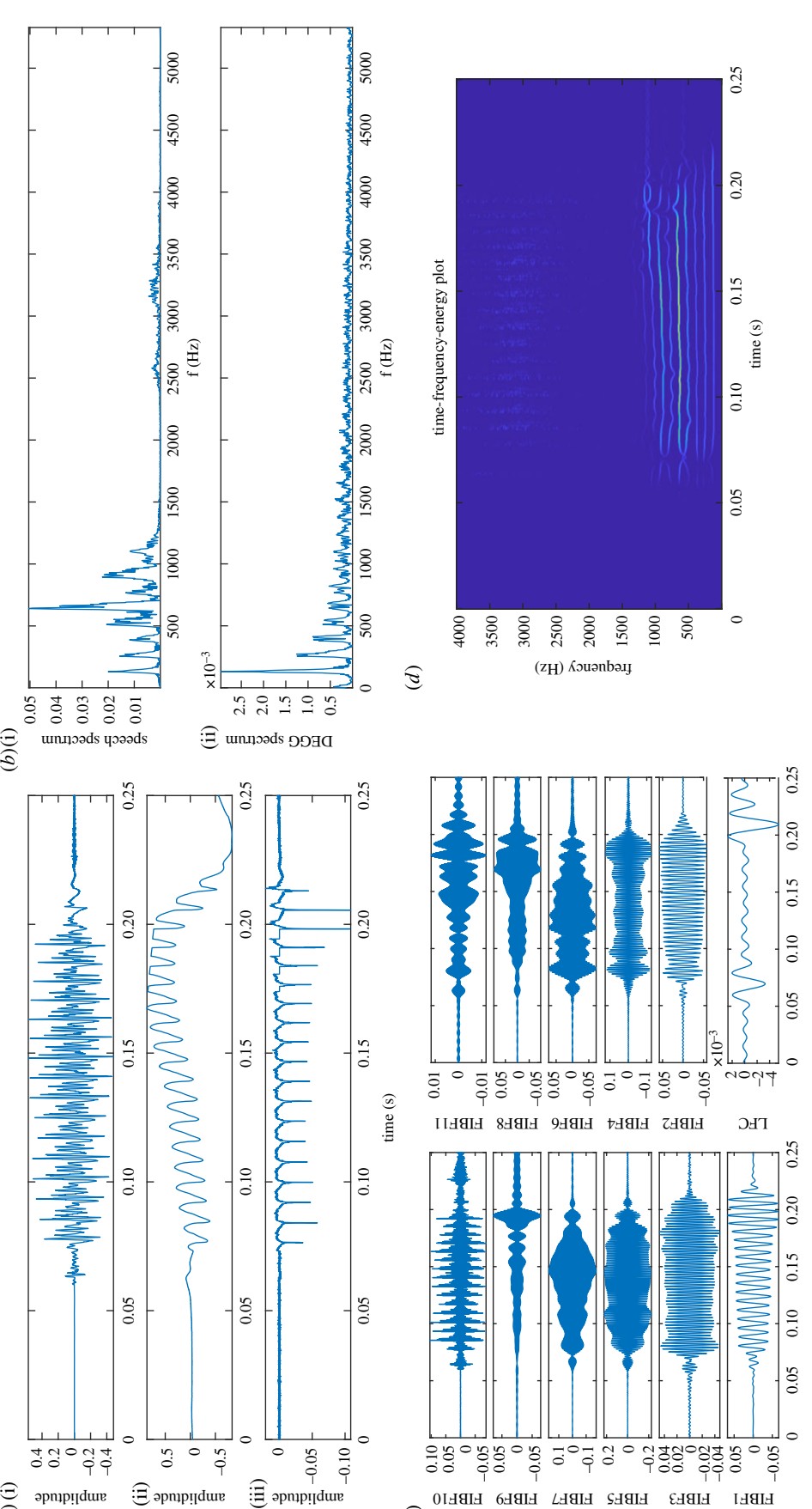

**Figure 4.** Analysis of speech signal: (*a*) a segment of speech (i), EGG (ii) and DEGG (iii) signals, (*b*) spectrum of speech (i), EGG (ii) and DEGG (iii) signals, (*c*) set of FIBFs obtained by the proposed DCT-based FDM and (*d*) TFE representation by proposed method.

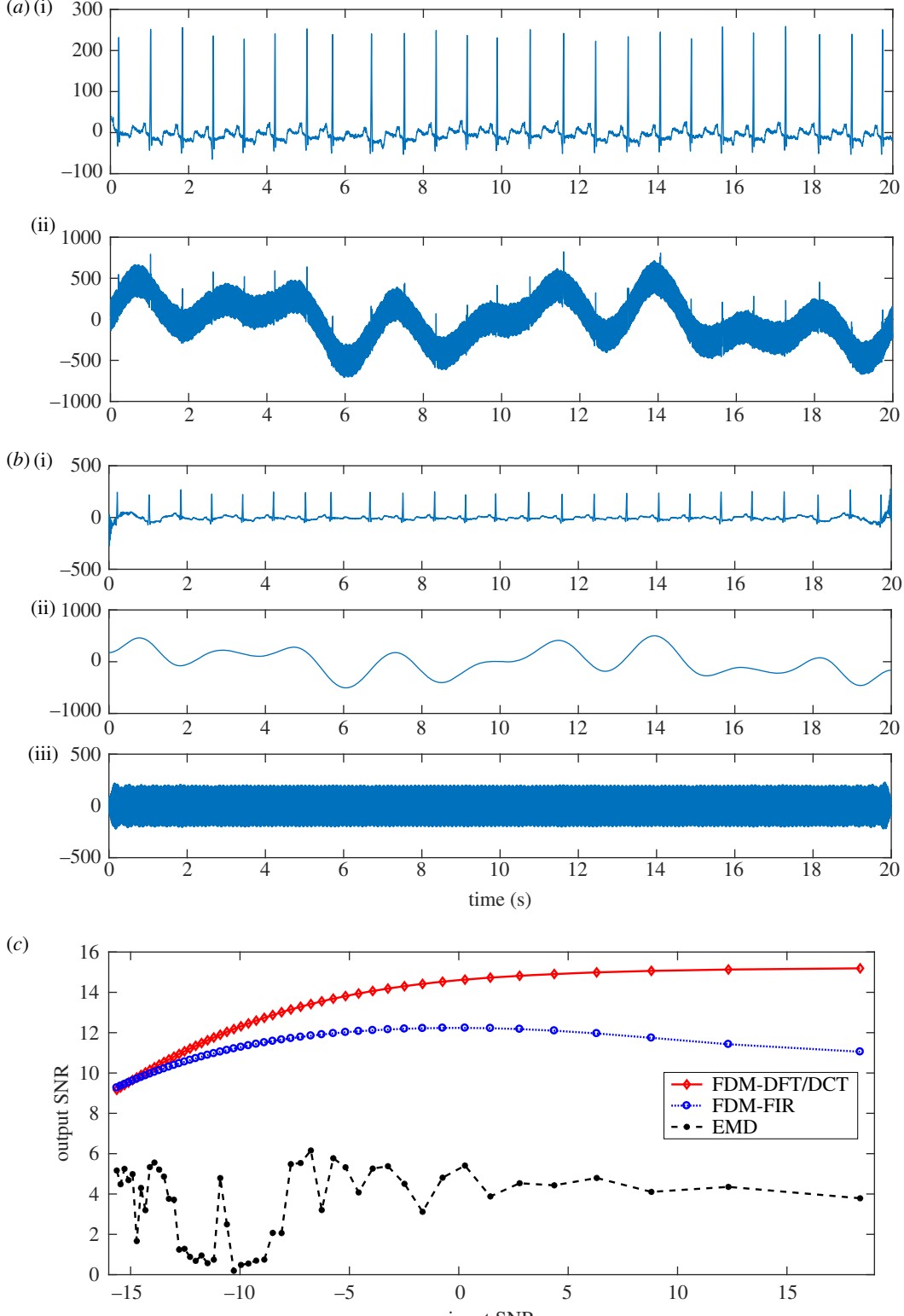

**Figure 5.** Noise removal from ECG signal using the proposed method: (*a*) a segment of clean ECG signal (i), ECG signal corrupted (SNR ≈ −18.4 dB) with baseline wander and power-line interference (ii), (*b*) ECG signal after noise removal (i), estimated baseline wander (ii) and power-line interference (iii), (*c*) a comparison of the EMD algorithm and FDM using input and output SNR criterion for application of noise removal from ECG signal.

$SNR_o = 14.02$ dB for $SNR_i = 5$ dB; and $SNR_o = 8.62$ dB for $SNR_i = −10$ dB. The proposed FDM−DCT provides better results, as shown in figure 5*c*, as $SNR_o = 14.99$ dB for $SNR_i = 5$ dB and $SNR_o = 12.32$ dB for $SNR_i = −10$ dB.

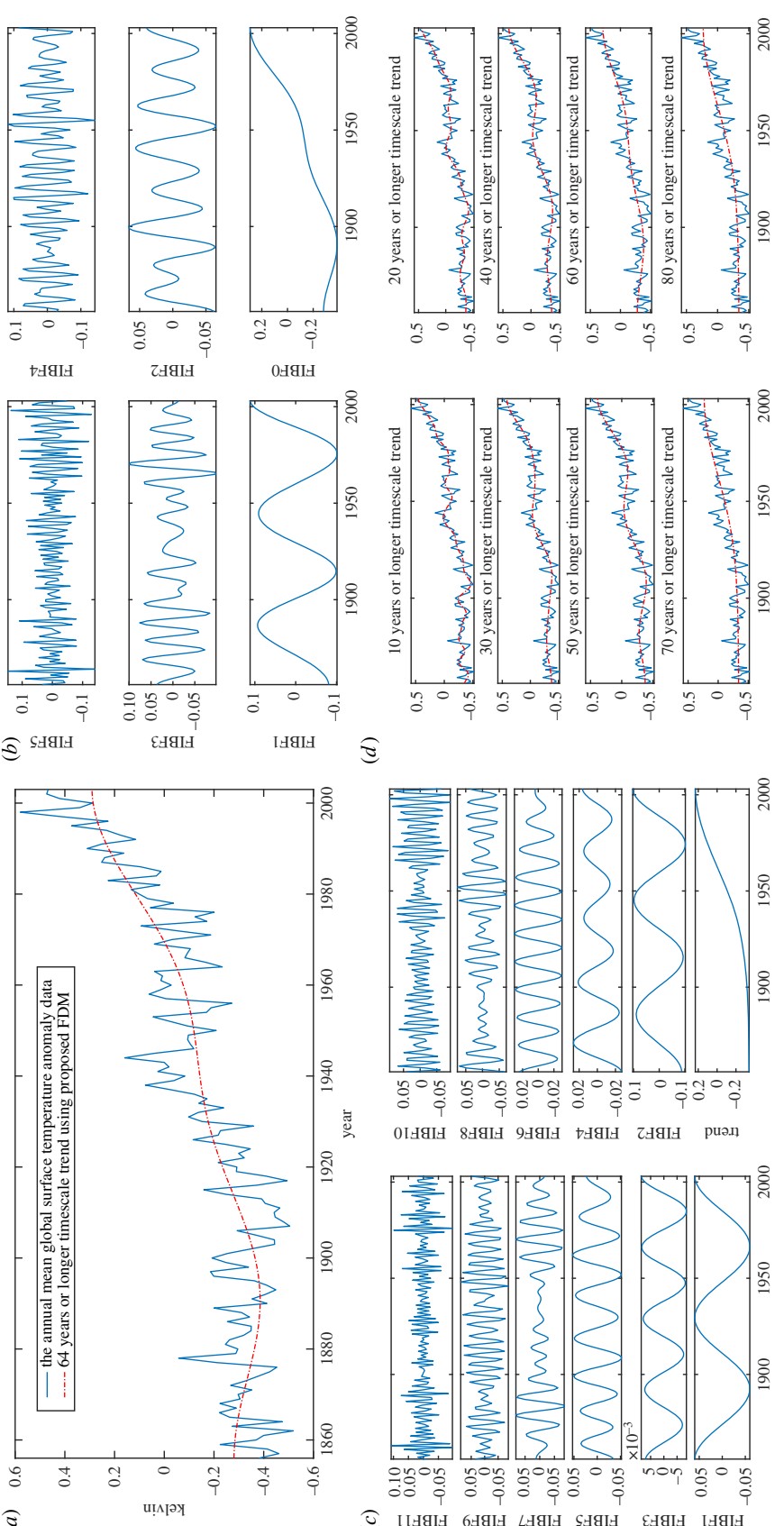

**Figure 6.** The GSTA data analysis using the proposed FDM: (*a*) GSTA data (blue solid) and its trend (red dashed) obtained by dividing data into six dyadic frequency bands, and corresponding dyadic FIBFs (*b*), (*c*) FIBFs obtained by dividing data into 12 non-dyadic frequency bands, and (*d*) GSTA data (blue solid) and trends (red dashed) in various timescales of 10–80 years or longer.

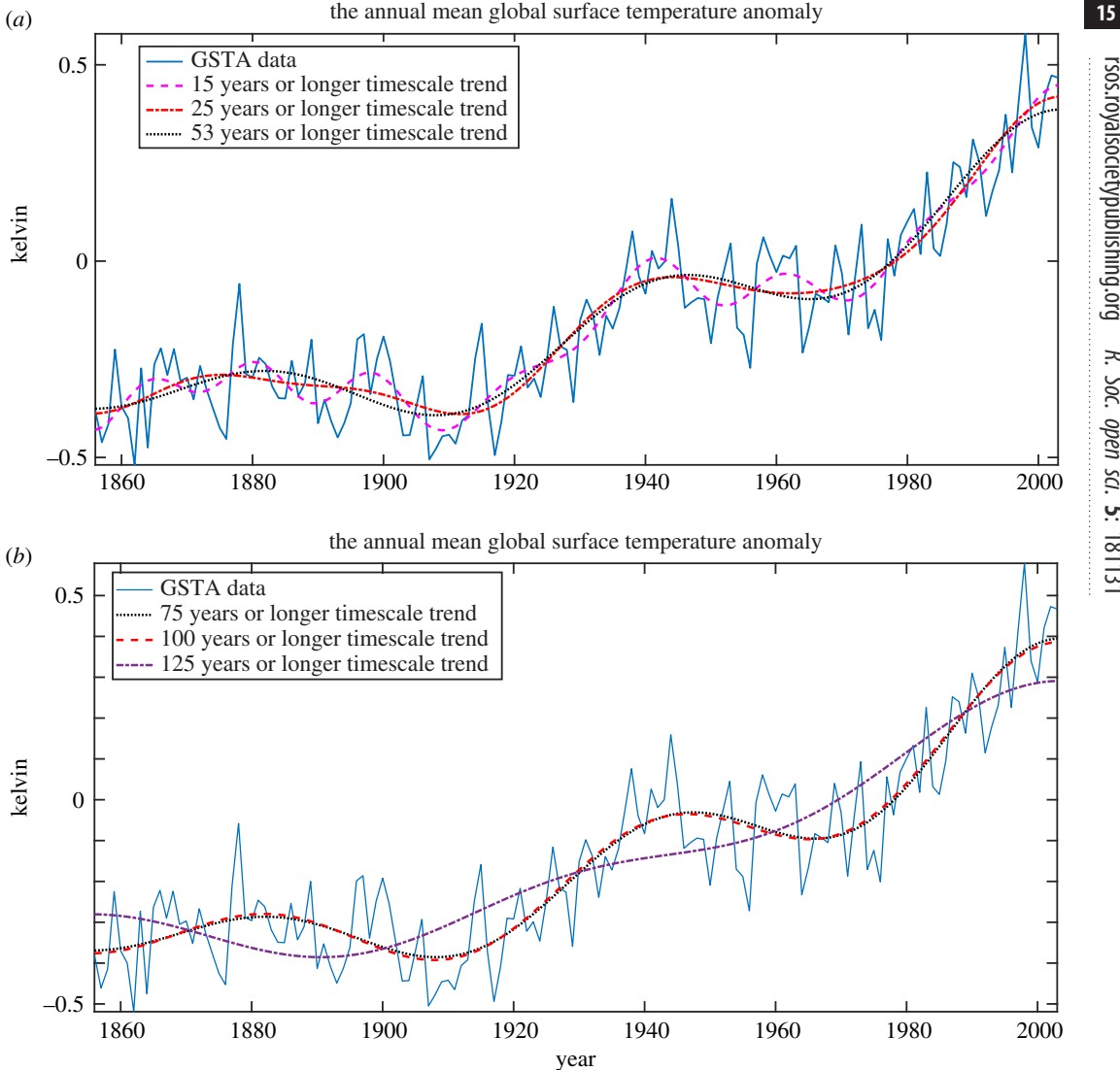

**Figure 7.** Trend analysis of GSTA data using the proposed FDM: (*a*) GSTA data, and its trend in 15, 25 and 53 years or longer timescale, (*b*) GSTA data, and its trend in 75, 100 and 125 years or longer timescale.

## 4.5. Trend and variability estimation from a nonlinear and non-stationary time series

Estimating trend and performing detrending operations are important steps in numerous applications, e.g. in climatic data analyses, the trend is one of the most critical parameters [65]; in spectral analysis and in estimating the correlation function, it is necessary to remove the trend from the data to obtain meaningful results [65]. Thus, in signal and other data analysis, it is crucial to estimate the trend, detrend the data and obtain variability of the time series at any desired timescales of interest.

Here, we consider the global surface temperature anomaly (GSTA) data (of 148 Years from 1856 to 2003, publicly available at http://rcada.ncu.edu.tw/research1_clip_ex.htm.)) analysis, its trend and variabilities obtained from the proposed method, as shown in figure 6: (*a*) GSTA data (blue solid), and its trend (red dashed, 64 years or longer timescale) obtained by dividing data into six dyadic frequency bands and (*b*) corresponding FIBFs, where FIBF5 shows (2–4) years timescale variations, FIBF4 (4–8), FIBF3 (8–16), FIBF2 (16–32), FIBF1 (32–64) and FIBF0 shows 64 years or longer timescale variations. Figure 6*c* was obtained by decomposing the data into 12 FIBFs of non-dyadic frequency bands: FIBF11 shows (2–3) years timescale variations, FIBF10 (3–4), FIBF9 (4–6), FIBF8 (6–8), FIBF7 (8–12), FIBF6 (12–16), FIBF5 (16–24), FIBF4 (24–32), FIBF3 (32–44), FIBF2, (44–64), FIBF1 (64–80 years) and trend shows 80 years or longer timescale variations of temperature.

To demonstrate the efficacy of the proposed FDM to estimate a trend and variability from data at any desired timescale, e.g. we estimated the trends and variabilities of GSTA data in various timescales as

rsos.royalsocietypublishing.org    R. Soc. open sci. **5**: 181131

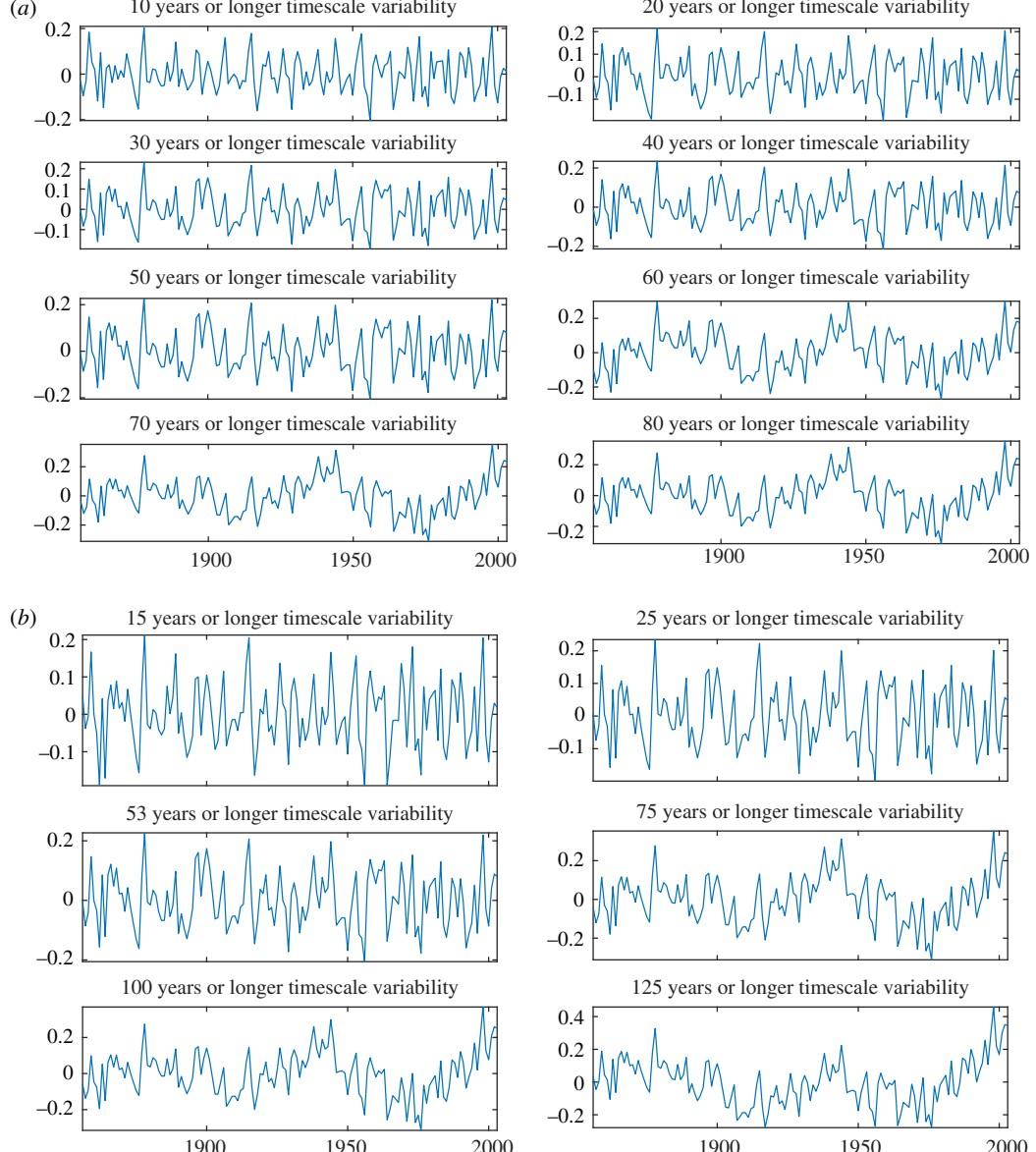

**Figure 8.** Variability analysis of GSTA data using the proposed FDM: (*a*) variabilities in multiple of 10 (i.e. 10−80) years or longer timescales corresponding to trends in figure 6*d* and (*b*) variabilities in 15, 25, 53, 75, 100 and 125 years or longer timescales.

shown in figures 6*d*, 7 and 8. Figure 6*d* shows GSTA data (blue solid), and trends (red dashed) in multiple of 10 (i.e. 10−80) years or longer timescale variations of temperature. Figure 7 presents: (*a*) GSTA data, and its trend in 15, 25 and 53 years or longer timescale and (*b*) GSTA data, and its trend in 75, 100 and 125 years or longer timescale.

The variabilities of the annual GSTA data corresponding to various trends (i.e. data minus corresponding trends) with their timescales are given in figure 8: (*a*) variabilities corresponding to trends in figure 6*d* and (*b*) variabilities corresponding to trends in figure 7. From these figures, it is clear that the variabilities up to 53 years or longer timescale are similar and do not contain any low-frequency component (i.e. there is no mode-mixing). However, the variabilities from 54 years (experimentally found accurately 54 years as shown in figure 9) or longer timescale are different from up to 53 years and also contain low-frequency components (i.e. there is mode-mixing).

The trend and variability analysis of this climate data is also performed by the EMD algorithm in [65]. The EMD algorithm decomposes the climate data into a set of IMFs which have overlapping frequency spectra and their cut-off frequencies are also not under control, therefore the trend and variability of data at desired timescales with clear demarcation cannot be obtained. For example, we have shown that there is a significant change in the trend and variability of data in just 1 year difference of timescale (observe

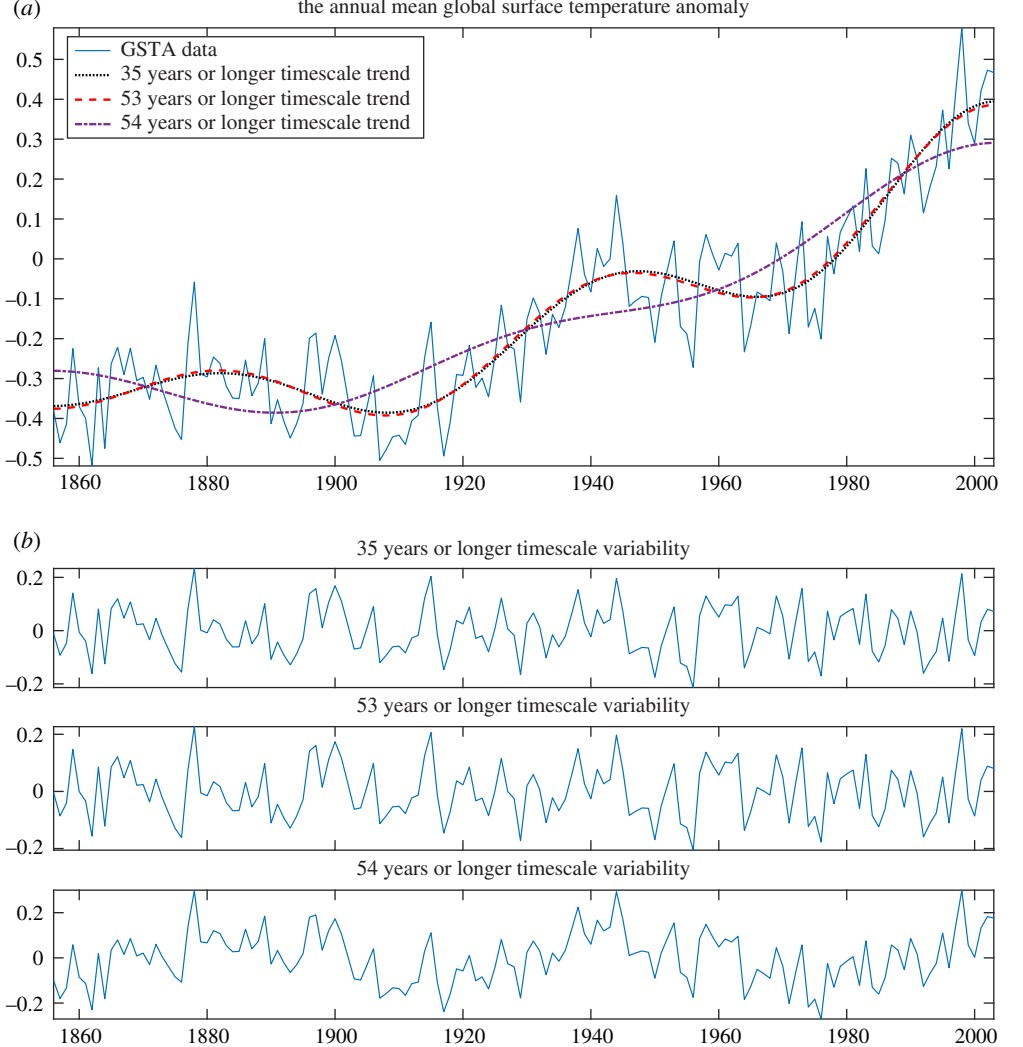

**Figure 9.** Trend and variability analysis of GSTA data using the proposed FDM: (*a*) GSTA data, and its trend in 35, 53 and 54 years or longer timescale and (*b*) corresponding variabilities.

change in 53 years or longer timescale to 54 years or longer timescale in figure 9). This kind of fine control along with determination of trend and variability of data in desired timescale with clear demarcation can be obtained by the proposed method, which is not achievable by the EMD and its related algorithms.

## 4.6. Seismic signal analysis

An earthquake time series is nonlinear and non-stationary in nature, and in this example, we consider the Elcentro Earthquake data. The Elcentro Earthquake time series (which is sampled at $F_s = 50$ Hz) has been downloaded from http://www.vibrationdata.com/elcentro.htm and is shown in figure 10*a*(i). The most critical frequency range that matters in the structural design is less than 10 Hz, and the Fourier power spectral density (PSD), figure 10*a*(ii), shows that almost all the energy in this earthquake time series is present within 10 Hz. Figure 10*b* presents TFE distributions by the DCT-based FDM method without decomposition, (*c*) presents FIBFs obtained by decomposing data into eight dyadic bands (FIBF0 (0–0.1953) Hz, FIBF1 (0.1953–0.390), FIBF2 (0.390–0.78125), FIBF3 (0.78125–1.5625), FIBF4 (1.5625–3.125), FIBF5 (3.125–6.25), FIBF6 (6.25–12.5), FIBF7 (12.5–25) Hz), (*d*) presents 10 equal energy FIBFs (FIBF0 (0–927.7344e-003) Hz, FIBF0 (927.7344e-003–1.1841), FIBF2 (1.1841–1.5015), FIBF3 (1.5015–1.8433), FIBF4 (1.8433–2.1606), FIBF5 (2.1606–2.6733), FIBF6 (2.6733–3.7109), FIBF7 (3.7109–4.7119), FIBF8 (4.7119–6.8237), FIBF9 (6.8237–25) Hz), (*e*) presents TFE distribution corresponding to eight dyadic bands and (*f*) presents TFE distribution corresponding to 10 equal energy bands. The obtained TFE distributions indicate that the maximum energy in the signal is present around 2 s and 1.7 Hz. These

rsos.royalsocietypublishing.org R. Soc. open sci. **5**: 181131

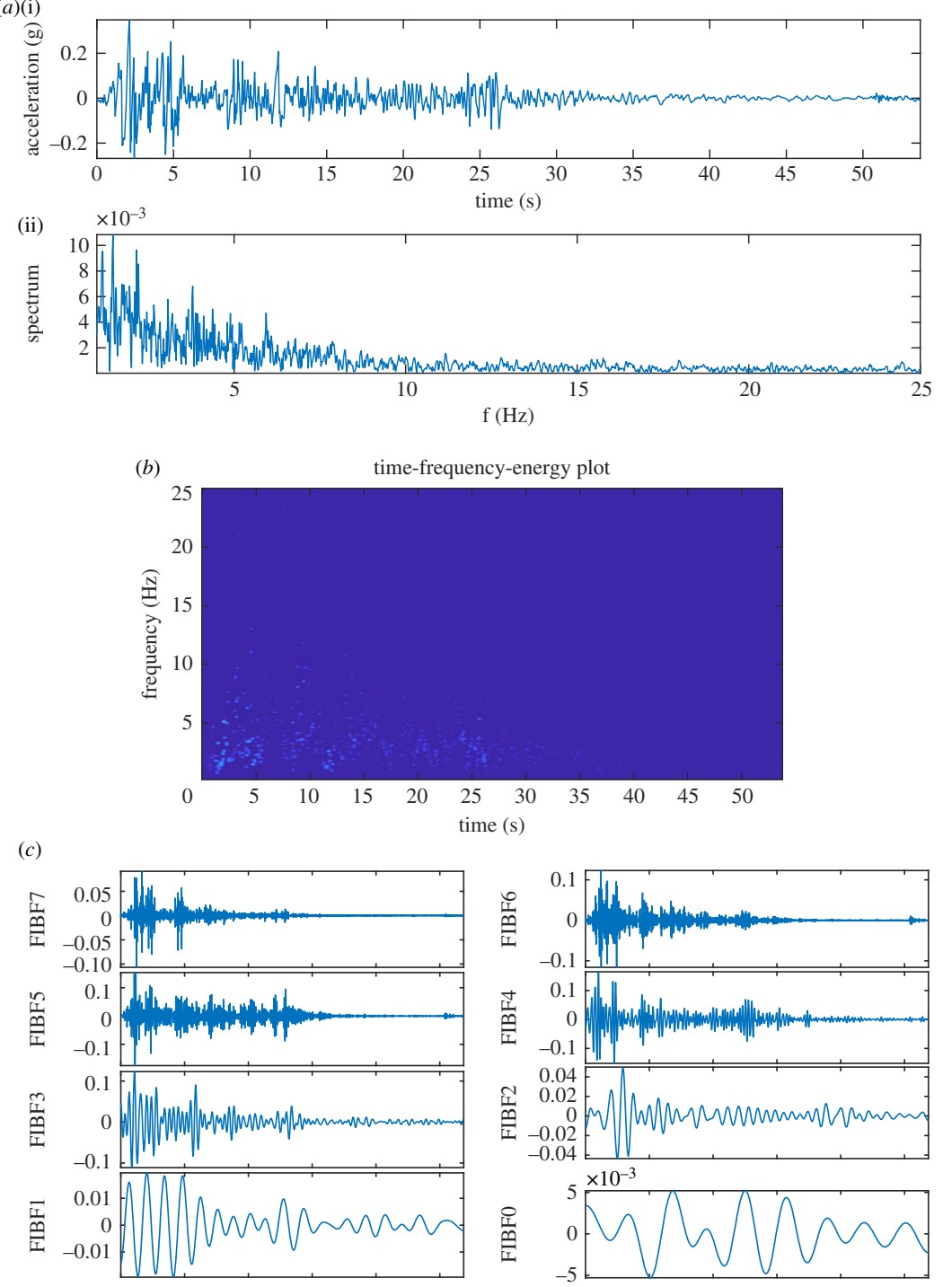

**Figure 10.** The Elcentro Earthquake (18 May 1940) data analysis using the proposed FDM: (*a*) North–South component EQ data (i), and its Fourier spectrum (ii), (*b*) TFE distribution without any decomposition, (*c*) FIBFs (eight dyadic frequency bands), (*d*) FIBFs (10 equal-energy frequency bands), (*e*) TFE corresponding to eight dyadic frequency bands and (*f*) TFE corresponding to 10 equal-energy frequency bands.

FIBFs and TFE distributions provide details of how the different waves arrive from the epical centre to the recording station, for example, the compression waves of small amplitude and higher frequency range 12–25 Hz (e.g. FIBF7 of figure 10*c*), the shear and surface waves of strongest amplitude and lower frequency range of below 12 Hz (e.g. FIBF6, FIBF5, FIBF4 and FIBF3 of figure 10*c*) which create most of the damage in structure, and other body shear waves of lowest amplitude and frequency range of below 1 Hz (e.g. FIBF2, FIBF1, and FIBF0 of figure 10*c*) which are present over the full duration of the time series.

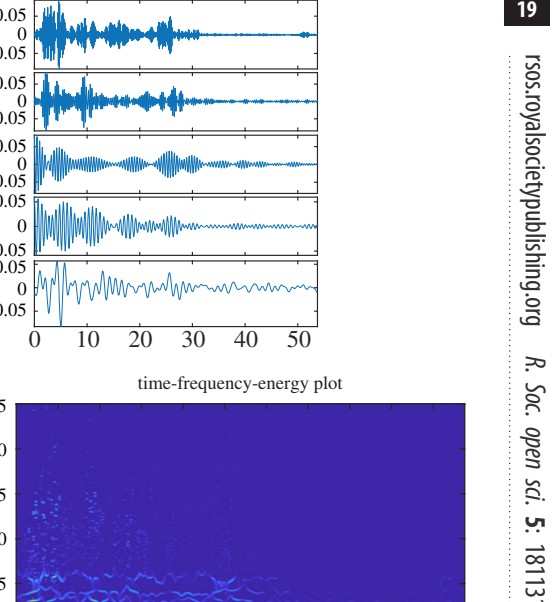

rsos.royalsocietypublishing.org    R. Soc. open sci. **5**: 181131

Figure 10. (*Continued.*)

## 4.7. Gravitational wave event GW150914 analysis: noise removal, IF and TFE estimation

Gravitational waves (GWs), predicted in 1916 by Albert Einstein, are ripples in the space–time continuum that travel outward from a source at the speed of light, and carry with them information about their source of origin. The GW event GW150914, produced by a binary black hole merger nearly 1.3 billion light years away [50], marks one of the greatest scientific discoveries in the history of human life. In this example, using the proposed method, we consider the noise removal, IF estimation and TFE representation of the GW event data which is publicly available at https://losc.ligo.org/events/GW150914/ (sampling rate $F_s = 16384$ Hz). The GW signal sweeps upwards in frequency from 35 to 250 Hz, and amplitude strain increases to a peak GW strain of $1.0 \times 10^{-21}$ [50]. The noise removal and an accurate IF estimation of the GW data is important because IF is the basis to estimate many parameters such as velocity, separation, luminosity distance, primary mass, secondary mass, chirp mass, total mass and effective spin of binary black hole merger [50]. Figure 11 shows (*a*) the GW H1 strain data GW150914 (i) observed at the Laser Interferometer Gravitational-Wave Observatory (LIGO) Hanford, which is heavily corrupted with noise, the Fourier spectrum (ii) of the GW data, which is not able to capture the non-stationarity (i.e. upwards sweep in the frequency and amplitude) present in the data, (*b*) the TFE representation of the GW H1 data without decomposition by proposed method, which shows the signal frequency is increasing with time but having lots of unnecessary fluctuations in frequency due to noise, and (*c*) decomposition of data into a set of six FIBFs (FIBF1 (25–60), FIBF2 (60–100), FIBF3 (100–200), FIBF4 (200–300), FIBF5 (300–350), FIBF6 (350–8192) Hz) and a low-frequency component (LFC) of band 0–25 Hz, (*d*) FW1−FW5 are obtained by multiplying the time domain Gaussian window with corresponding FIBFs (FIBF1–FIBF5), and the reconstructed GW (RGW) is obtained by summation of FW1−FW5 components. In the reconstruction of wave, the LFC and FIBF6 have been ignored as they are out of band noises present in the GW H1 data.

The further analysis, comparison, residue and TFE estimation of GW event GW150914 H1 strain data (captured at LIGO Hanford) using the proposed FDM are shown in figure 12: (*a*) GW H1 strain data ((i) red dashed line), proposed reconstruction ((i) blue solid line), and estimated residue component (ii), (*b*) numerical relativity (NR) data [50] ((i) red dashed line), and proposed reconstruction ((i) blue solid line), and difference between NR and reconstructed data (ii); TFE estimates of the NR data and the reconstructed data are shown in (*c*) and (*d*), respectively. The very same analysis of the GW event GW150914 L1 strain data (captured at LIGO Livingston) using the proposed method is shown in figure 13.

rsos.royalsocietypublishing.org    R. Soc. open sci. **5**: 181131

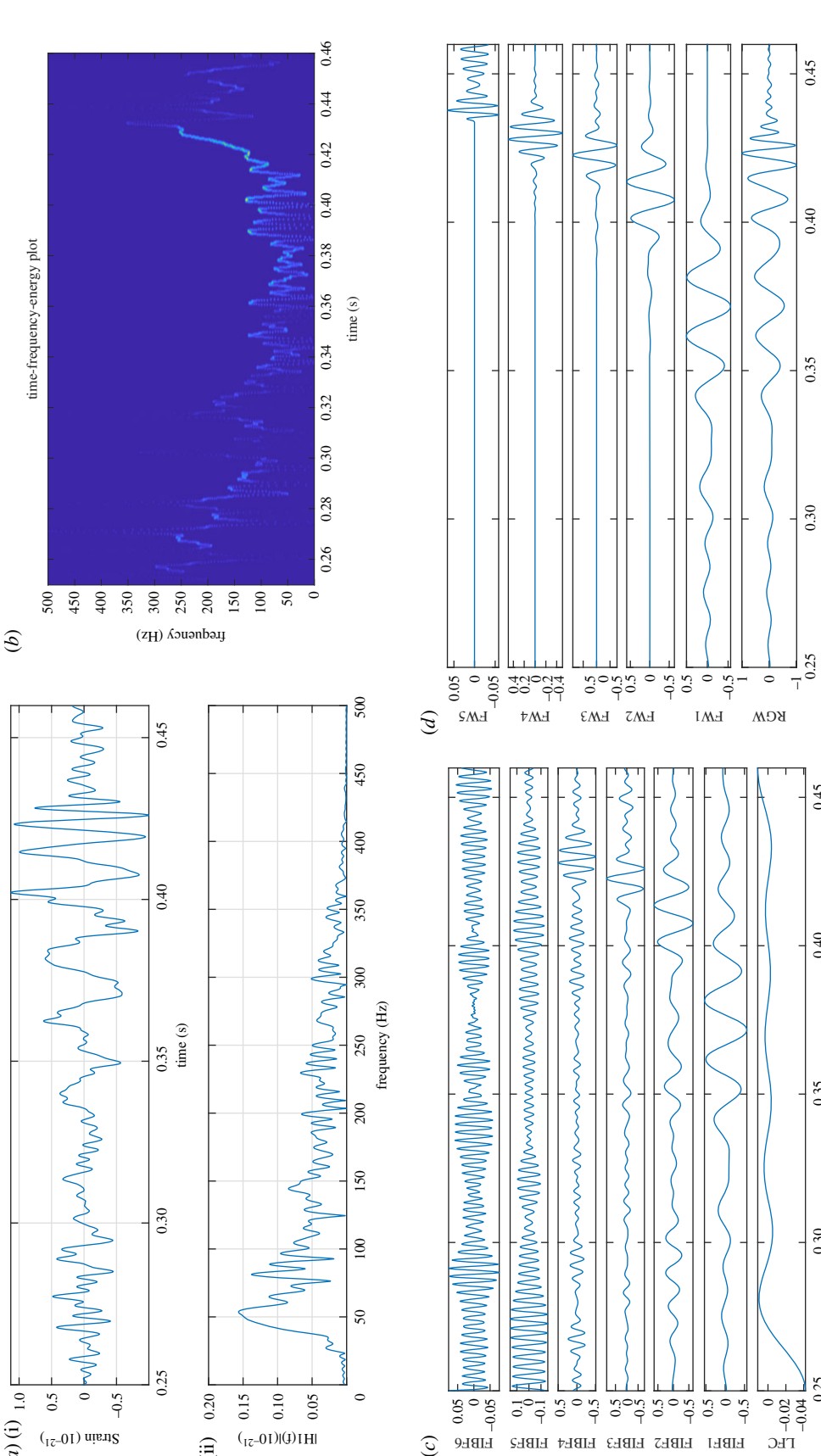

**Figure 11.** The GW event GW150914 H1 strain data (captured at LIGO Hanford) analysis using the proposed FDM: (*a*) The GW H1 strain data [50] (i), and its Fourier spectrum (ii), (*b*) TFE estimates without any decomposition, (*c*) obtained FIBF1−FIBF6 and low-frequency component (LFC), (*d*) results (FW1−FW5) obtained by multiplication of FIBF1−FIBF6 with corresponding Gaussian windows, and reconstructed gravitational wave (RGW) by sum of FW1−FW5.

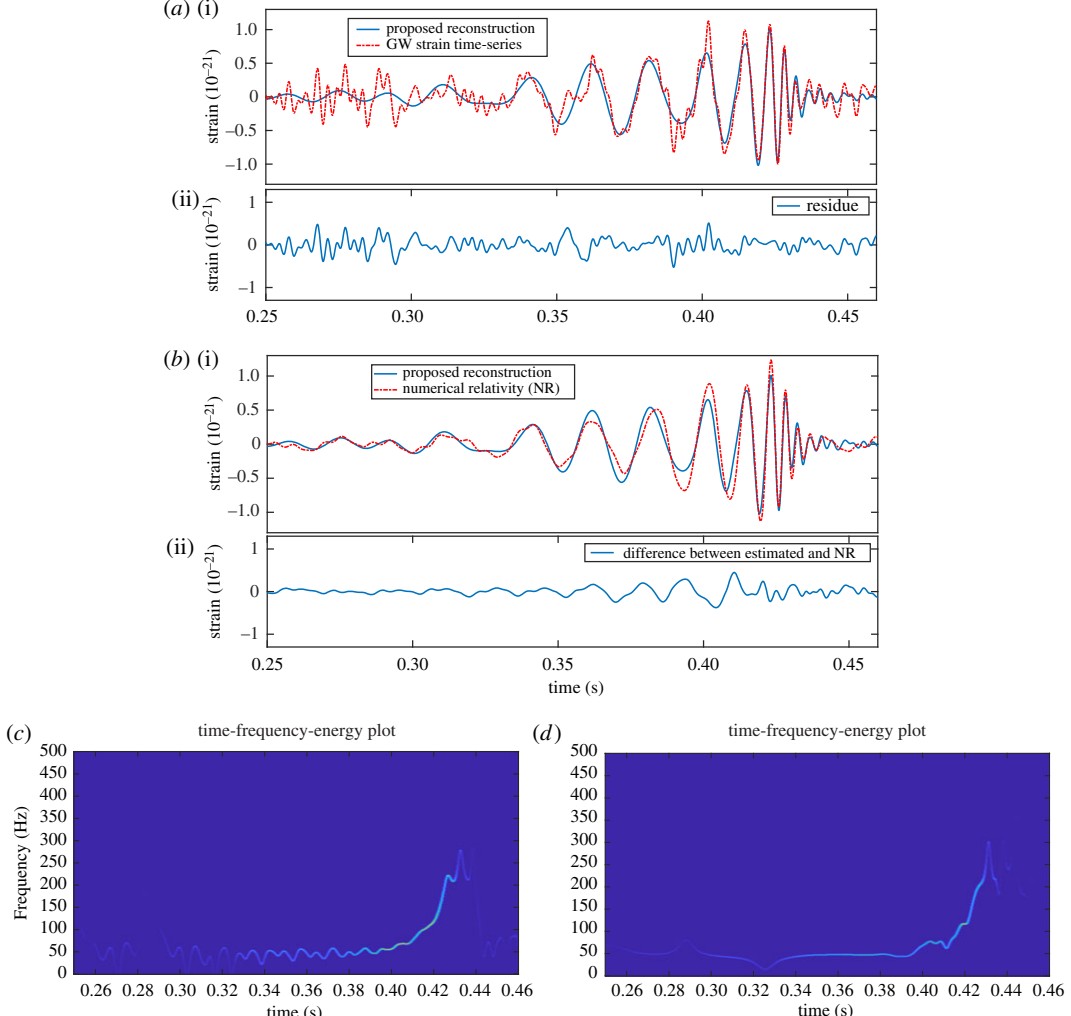

**Figure 12.** The GW event GW150914 H1 strain data analysis using the proposed FDM: (*a*) GW H1 strain data ((i) red dashed line), and proposed reconstruction ((i) blue solid line), and residue component (ii). (*b*) Numerical relativity (NR) data ((i) red dashed line), and proposed reconstruction ((i) blue solid line), and difference between NR and reconstructed data (ii), (*c*) TFE estimates of the NR data and (*d*) TFE estimates of the reconstructed data.

These examples clearly demonstrate the efficacy of the proposed method for the analysis of real-life non-stationary signals such as speech (§4.3), ECG (§4.4), climate (§4.5), seismic (§4.6) and gravitational (§4.7) time-series. This study is aimed to complement the current nonlinear and non-stationary data processing methods with the addition of the FDM, which is based on the DCT, discrete FCQT and zero-phase filter approach using GAS and FSAS representations.

# 5. Conclusion

The HT, to obtain quadrature component and GAS representation, is most well known, widely used and a key mathematical representation for modelling and analysis of signals to reveal underlying physical phenomenon in various applications. The fundamental contributions and conceptual innovations of this study are introduction of the new discrete FCQTs and discrete FSQTs, designated as FQTs, as effective alternatives to the HT. Using these FQTs, we proposed Fourier quadrature analytic signal (FQAS) representations, as coherent alternatives to the GAS, with following properties: (1) real part of FQAS is the original signal and imaginary part is the FCQT of the real part, (2) imaginary part of FQAS is the original signal and real part is the FSQT of the imaginary part, (3) like the GAS, Fourier spectrum of the FQAS has only positive frequencies; however, unlike the GAS, the real and imaginary parts of the proposed FQAS representations are not orthogonal to each other. Moreover,

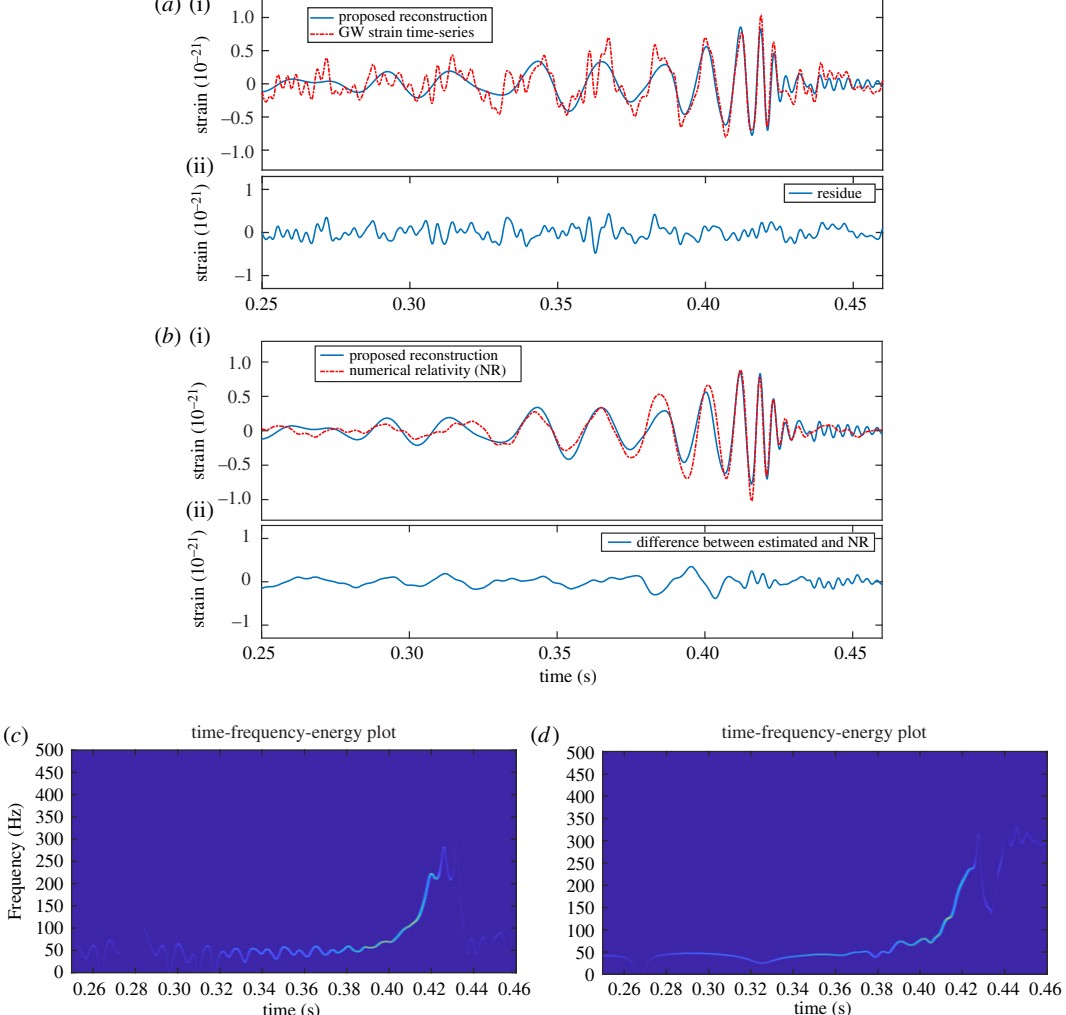

**Figure 13.** The GW event GW150914 L1 strain data (captured at LIGO Livingston) analysis using the proposed FDM: (*a*) GW L1 strain data ((i) red dashed line), and proposed reconstruction ((i) blue solid line), and residue component (ii). (*b*) Numerical relativity (NR) data ((i) red dashed line), and proposed reconstruction ((i) blue solid line), and difference between NR and reconstructed data (ii), (*c*) TFE estimates of the NR data, and (*d*) TFE estimates of the reconstructed data.

continuous-time FQTs, FQAS representations, and two-dimensional extension of these formulations are also presented.

The Fourier theory and ZPF-based FDM is an adaptive data analysis approach to decompose a signal into a set of small number of Fourier intrinsic band functions (FIBFs). This study also proposed a new formulation of the FDM using the discrete cosine transform and the discrete cosine quadrature transform with the GAS and FQAS representations, and demonstrated its efficacy for analysis of nonlinear and non-stationary simulated as well as real-lifetime series such as instantaneous fundamental frequency estimation from unit sample sequence, chirp and speech signals, baseline wander and power-line interference removal from ECG data, trend and variability estimation and analysis of climate data, seismic time series analysis, and finally noise removal, IF and TFE estimation from the gravitational wave event GW150914 strain data.

Many time-frequency representation approaches such as the EMD algorithms, FDM and VMD are primarily based on the HT and GAS representation which are well known and widely used in numerous applications. This study proposes 16 FQTs as alternatives of the HT, 16 FQAS representations as alternatives of the GAS, and presents detailed simulation results using DCT-2-based FQT and FQAS, which generates better results (e.g. in terms of accurate instantaneous frequency estimation, higher output SNR, trend and variability estimation in desired timescale) in many but limited applications considered in this work. Moreover, the proposed novel methodology provides alternative ways of time-series representation and analysis, advances the existing FDM, which is useful many applications, and therefore, these transforms and corresponding analytic signal

representations are required. Finally, as the DFT and DCTs are widely used tools in numerous different sets of applications, future direction of research would be to explore proposed FQTs and corresponding analytic signal representations in various applications.

Data accessibility. The FDM Matlab code is publicly available for download at the Dryad Digital Repository (doi:10.5061/dryad.jc21t36) [58] and https://www.researchgate.net/publication/326294577_MATLABCodeOfFDM_DCT_DFT_FIR_FSASJuly2018. All data used in this study are publicly available and links are included in the references of paper.

Competing interests. I declare I have no competing interests.

Funding. I thank SEAS, ECE Department, Bennett University, Greater NOIDA, UP, India for providing Royal Society *Open Science* article processing charges for this research.

Acknowledgements. I thank SEAS, Bennett University, Greater NOIDA for providing research facilities to carry out this research. I also thank the anonymous reviewers for their valuable review comments and suggestions which helped in improving the quality of the manuscript.

# Appendix A. Continuous-time Fourier quadrature transforms

The Fourier cosine transform (FCT) and inverse FCT (IFCT) pairs, of a signal, are defined as

$$X_c(\omega) = \sqrt{\frac{2}{\pi}} \int_0^\infty x(t) \cos(\omega t) \, dt, \qquad \omega \geq 0$$

and

$$x(t) = \sqrt{\frac{2}{\pi}} \int_0^\infty X_c(\omega) \cos(\omega t) \, d\omega, \quad t \geq 0, \qquad (A\,1)$$

subject to the existence of the integrals, i.e. $x(t)$ is absolutely integrable ($\int_0^\infty |x(t)| \, dt < \infty$) and its derivative $x'(t)$ is piece-wise continuous in each bounded subinterval of $[0, \infty)$.

We hereby define the FCQT, $\tilde{x}_c(t)$, using the FCT of signal of $x(t)$ as

$$\tilde{x}_c(t) = \sqrt{\frac{2}{\pi}} \int_0^\infty X_c(\omega) \sin(\omega t) \, d\omega = \frac{2}{\pi} \int_0^\infty \left[ \int_0^\infty x(\tau) \cos(\omega \tau) \, d\tau \right] \sin(\omega t) \, d\omega. \qquad (A\,2)$$

From (A 2), we obtain

$$\tilde{X}_c(\omega) = \sqrt{\frac{2}{\pi}} \int_0^\infty \tilde{x}_c(t) \sin(\omega t) \, dt, \qquad (A\,3)$$

where

$$\tilde{X}_c(\omega) = \begin{cases} 0, & \omega = 0, \\ X_c(\omega), & \omega > 0. \end{cases} \qquad (A\,4)$$

The proposed FSAS, using the FCQT, is defined as

$$\tilde{z}_c(t) = x(t) + j\tilde{x}_c(t) = \sqrt{\frac{2}{\pi}} \int_0^\infty X_c(\omega) \exp(j\omega t) \, d\omega, \qquad (A\,5)$$

where real part is the original signal and imaginary part is the FQT of real part.

The Fourier sine transform (FST) and inverse FST (IFST) pairs, of a signal, are defined as

$$X_s(\omega) = \sqrt{\frac{2}{\pi}} \int_0^\infty x(t) \sin(\omega t) \, dt$$

and

$$x(t) = \sqrt{\frac{2}{\pi}} \int_0^\infty X_s(\omega) \sin(\omega t) \, d\omega, \qquad (A\,6)$$

subject to the existence of the integrals. We hereby define the FSQT, $\tilde{x}_s(t)$, using the FST of signal of $x(t)$ as

$$\tilde{x}_s(t) = \sqrt{\frac{2}{\pi}} \int_0^\infty X_s(\omega) \cos(\omega t) \, d\omega = \frac{2}{\pi} \int_0^\infty \left[ \int_0^\infty x(\tau) \sin(\omega \tau) \, d\tau \right] \cos(\omega t) \, d\omega. \qquad (A\,7)$$

From (A 7), we can write

$$\tilde{X}_s(\omega) = \sqrt{\frac{2}{\pi}} \int_0^\infty \tilde{x}_s(t) \cos(\omega t) \, dt, \qquad (A\,8)$$

and one can observe that both representations, defined as FST of $x(t)$ in (A 6) and FCT of $\tilde{x}_s(t)$ in (A 8), are the same for all frequencies, i.e. $X_s(\omega) = \tilde{X}_s(\omega)$.

The proposed FSAS, using the FSQT, is defined as

$$\tilde{z}_s(t) = \tilde{x}_s(t) + jx(t) = \sqrt{\frac{2}{\pi}} \int_0^\infty X_s(\omega) \exp(j\omega t)\, d\omega, \tag{A 9}$$

where imaginary part is the original signal and real part is the FQT of imaginary part. The FQTs, presented in (A 2) and (A 7), are different from the HT by definition itself. The proposed FSAS representations, defined in (A 5) and (A 9), are effective alternatives to the GAS representation which is used in various applications such as envelop detection, IF estimation, time-frequency-energy representation and analysis of nonlinear and non-stationary data.

# Appendix B. The two-dimensional DCT and corresponding FSAS representations

The two-dimensional DCT-2 of a sequence, $x[n, m]$, is defined as [57]

$$X_2[k, l] = \frac{2}{\sqrt{MN}} \sum_{m=0}^{M-1} \sum_{n=0}^{N-1} \sigma_k \sigma_l x[m, n] \cos\left(\frac{\pi k(2m+1)}{2M}\right) \cos\left(\frac{\pi l(2n+1)}{2N}\right), \tag{B 1}$$

and IDCT is obtained by

$$x[m, n] = \frac{2}{\sqrt{MN}} \sum_{k=0}^{M-1} \sum_{l=0}^{N-1} \sigma_k \sigma_l X_2[k, l] \cos\left(\frac{\pi k(2m+1)}{2M}\right) \cos\left(\frac{\pi l(2n+1)}{2N}\right). \tag{B 2}$$

Using the relation, $\cos(\alpha)\cos(\beta) = \frac{1}{2}[\cos(\alpha + \beta) + \cos(\alpha - \beta)]$, we obtain the two-dimensional FSAS as

$$\tilde{z}[m, n] = \frac{1}{\sqrt{MN}} \sum_{k=0}^{M-1} \sum_{l=0}^{N-1} \sigma_k \sigma_l X_2[k, l] \left[ \exp\left(\frac{j\pi k(2m+1)}{2M} + \frac{j\pi l(2n+1)}{2N}\right) \right.$$
$$\left. + \exp\left(\frac{j\pi k(2m+1)}{2M} - \frac{j\pi l(2n+1)}{2N}\right) \right] = x[m, n] + j\tilde{x}[m, n], \tag{B 3}$$

where real part of (B 3) is the two-dimensional original signal and imaginary of it is the two-dimensional quadrature component of real part, which can be written as, using $\sin(\alpha)\cos(\beta) = \frac{1}{2}[\sin(\alpha + \beta) + \sin(\alpha - \beta)]$,

$$\tilde{x}[m, n] = \frac{2}{\sqrt{MN}} \sum_{k=0}^{M-1} \sum_{l=0}^{N-1} \sigma_k \sigma_l X_2[k, l] \sin\left(\frac{\pi k(2m+1)}{2M}\right) \cos\left(\frac{\pi l(2n+1)}{2N}\right). \tag{B 4}$$

From (B 4), one can obtain

$$\tilde{X}_2[k, l] = \frac{2}{\sqrt{MN}} \sum_{m=0}^{M-1} \sum_{n=0}^{N-1} \sigma_k \sigma_l \tilde{x}[m, n] \sin\left(\frac{\pi k(2m+1)}{2M}\right) \cos\left(\frac{\pi l(2n+1)}{2N}\right), \tag{B 5}$$

where

$$\tilde{X}_2[k, l] = \begin{cases} 0, & k = 0, & 0 \le l \le N-1, \\ X_2[k, l], & 1 \le k \le M-1, & 0 \le l \le N-1. \end{cases} \tag{B 6}$$

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
