## [Reviewer comments · Royal Society Open Science]

Review History

RSOS-181131.R0 (Original submission)

Review form: Reviewer 1

Is the manuscript scientifically sound in its present form?

Yes

Are the interpretations and conclusions justified by the results?

Yes

Is the language acceptable?

Yes

Is it clear how to access all supporting data?

Yes

Do you have any ethical concerns with this paper?

No

Have you any concerns about statistical analyses in this paper?

No

Recommendation?

Accept with minor revision (please list in comments)

Comments to the Author(s)

The content is technically correct. However, it is not placed in proper perspective. Some comments are given below:

1. The names of the proposed group of transforms and the resulting analytic signals should be suggestive that they are derived from the sine and cosine transforms.
2. Many signals can have same spectrum, and we know the reason for this fact. Now, one signal is having sixteen different time-frequency distributions. What is the meaning of this revelation?
3. Some discussion on why we need the above transforms will be highly desirable.

Review form: Reviewer 2

Is the manuscript scientifically sound in its present form?

Yes

Are the interpretations and conclusions justified by the results?

Yes

Is the language acceptable?

Yes

Is it clear how to access all supporting data?

Yes

Do you have any ethical concerns with this paper?

No

Have you any concerns about statistical analyses in this paper?

No

Recommendation?

Major revision is needed (please make suggestions in comments)

Comments to the Author(s)

This reviewer feels that the manuscript is weak in terms of recent literature related to the research work presented in it. The below mentioned references should be included in the manuscript before its acceptance for publication in order to make it very useful for readers and to show the effectiveness of the proposed work with reference to existing methods.

Recently, a new technique for time-frequency analysis which is based on improved eigenvalue decomposition and Wigner-Ville distribution has been proposed for cross-terms free time-

frequency representation for multicomponent non-stationary signals. Please see the following paper:

R.R. Sharma and R.B. Pachori, Improved eigenvalue decomposition-based approach for reducing cross-terms in Wigner-Ville distribution, *Circuits, Systems, and Signal Processing*, vol. 37, issue 08, pp. 3330-3350, August 2018.

Authors should discuss this method in the Introduction part of the paper. Also, explain how your proposed method is better than the above mentioned method can be explained.

In the other paper, authors have proposed a new method for time-frequency analysis using eigenvalue decomposition of Hankel matrix for complex signals. Please see below paper for more information.

R.R. Sharma and R.B. Pachori, Eigenvalue decomposition of Hankel matrix-based time-frequency representation of complex signals, *Circuits, Systems, and Signal Processing*, vol. 37, issue 08, pp. 3313-3329, August 2018.

The paper needs to cite this paper and should mention about the extension of the presented approach for time-frequency analysis for complex signals.

Recently, authors have proposed a new approach for time-frequency analysis using Fourier-Bessel series expansion based empirical wavelet transform and normalized Hilbert transform in the following paper:

A. Bhattacharyya, L. Singh, and R.B. Pachori, Fourier-Bessel series expansion based empirical wavelet transform for analysis of non-stationary signals, *Digital Signal Processing*, vol. 78, pp. 185-196, July 2018.

Authors should discuss this approach and developed method in the manuscript.

The other paper in literature develops a new method for time-frequency distribution based on improved eigenvalue decomposition and Hilbert transform.

R.R. Sharma and R.B. Pachori, Time-frequency representation using IEVDHM-HT with application to classification of epileptic EEG signals, *IET Science, Measurement & Technology*, vol. 12, issue 01, pp. 72-82, January 2018.

How the presented method is better than the suggested method in this paper? Please mention it in the paper.

In the literature, time-frequency analysis has been proposed for multivariate signals. Please see below paper.

A. Bhattacharyya and R.B. Pachori, A multivariate approach for patient specific EEG seizure detection using empirical wavelet transform, *IEEE Transactions on Biomedical Engineering*, vol. 64, no. 09, pp. 2003-2015, September 2017.

How the presented method can be extended for multivariate signals. This can be included as a future direction of this work with respect to the above mentioned paper.

In the literature, tunable-Q wavelet transform together with Wigner-Ville distribution has been studied for improving Wigner-Ville distribution based time-frequency analysis in the following paper:

R.B. Pachori and A. Nishad, Cross-terms reduction in Wigner-Ville distribution using tunable-Q wavelet transform, *Signal Processing*, vol. 120, pp. 288-304, March 2016.

How proposed method is different than this method for time-frequency analysis. Please mention in your paper.

Fourier-Bessel series expansion based time-order representation has been proposed for time-frequency analysis for speech signals in the following paper:

P. Jain and R.B. Pachori, Time-order representation based method for epoch detection from speech signals, *Journal of Intelligent Systems*, vol. 21, issue 1, pp. 79-95, February 2012.

How your method is expected to be better than this method? Please explain in the paper.

In another method of time-frequency analysis, Fourier-Bessel series expansion based component separation has been used together with Wigner-Ville distribution for improved time-frequency representation in the below paper.

R.B. Pachori and P. Sircar, A new technique to reduce cross terms in the Wigner distribution, *Digital Signal Processing*, vol. 17, no. 2, pp. 466-474, March 2007.

Please explain your method with respect to the presented method in the above mentioned paper.

Various applications of the developed method have been studied. Same applications have been reported in the below mentioned papers.

Recently, authors have proposed a new method for instantaneous fundamental frequency estimation using eigenvalue decomposition of the Hankel matrix in the following paper:

P. Jain and R.B. Pachori, Event-based method for instantaneous fundamental frequency estimation from voiced speech based on eigenvalue decomposition of Hankel matrix, *IEEE/ACM Transactions on Audio, Speech and Language Processing*, vol. 22, issue 10, pp. 1467-1482, October 2014.

Authors should also discuss the above mentioned paper in the interpretation of the obtained results using the proposed method.

The manuscript has also studied application of the proposed method for baseline wander removal. Recently, a new method based on the eigenvalue decomposition has been proposed for the same application. Please see the below reference.

R.R. Sharma and R.B. Pachori, Baseline wander and power line interference removal from ECG signals using eigenvalue decomposition, *Biomedical Signal Processing and Control*, vol. 45, pp. 33-49, August 2018.

Please discuss your obtained results for this application with reference to the above mentioned paper.

Review form: Reviewer 3 (Nizamettin Aydin)

Is the manuscript scientifically sound in its present form?

Yes

Are the interpretations and conclusions justified by the results?

Yes

Is the language acceptable?

Yes

Is it clear how to access all supporting data?

Yes

Do you have any ethical concerns with this paper?

No

Have you any concerns about statistical analyses in this paper?

No

Recommendation?

Accept as is

Comments to the Author(s)

The revised manuscript presents novel Fourier quadrature transforms and analytic signal representations for nonlinear and non-stationary time series analysis. The manuscript appears to be improved considerably by incorporating suggestions made by the reviewers. Although the language of the manuscript is in an acceptable level, it might be useful to get it read by a language expert as there are always some grammatical errors. For example, in page 9, line 4 from the bottom, delete "An" at the beginning of the sentence to start as "Estimation of...". Also, change "a" to "the" in "...is a most important..." to read as "...is the most important...".

Decision letter (RSOS-181131.R0)

18-Oct-2018

Dear Dr Singh,

The editors assigned to your paper ("Novel Fourier Quadrature Transforms and Analytic Signal Representations for Nonlinear and Non-stationary Time Series Analysis") have now received comments from reviewers. We would like you to revise your paper in accordance with the referee and Associate Editor suggestions which can be found below (not including confidential reports to the Editor). Please note this decision does not guarantee eventual acceptance.

Please submit a copy of your revised paper before 10-Nov-2018. Please note that the revision deadline will expire at 00.00am on this date. If we do not hear from you within this time then it will be assumed that the paper has been withdrawn. In exceptional circumstances, extensions may be possible if agreed with the Editorial Office in advance. We do not allow multiple rounds of revision so we urge you to make every effort to fully address all of the comments at this stage.

If deemed necessary by the Editors, your manuscript will be sent back to one or more of the original reviewers for assessment. If the original reviewers are not available, we may invite new reviewers.

- Data accessibility

If you wish to submit your supporting data or code to Dryad (<http://datadryad.org/>), or modify your current submission to dryad, please use the following link:
<http://datadryad.org/submit?journalID=RSOS&manu=RSOS-181131>

- Competing interests

- Authors' contributions

- Acknowledgements

- Funding statement

Please note that Royal Society Open Science charge article processing charges for all new submissions that are accepted for publication. Charges will also apply to papers transferred to Royal Society Open Science from other Royal Society Publishing journals, as well as papers submitted as part of our collaboration with the Royal Society of Chemistry (<http://rsos.royalsocietypublishing.org/chemistry>). If your manuscript is newly submitted and subsequently accepted for publication, you will be asked to pay the article processing charge, unless you request a waiver and this is approved by Royal Society Publishing. You can find out more about the charges at <http://rsos.royalsocietypublishing.org/page/charges>. Should you have any queries, please contact openscience@royalsociety.org.

on behalf of Prof. R. Kerry Rowe (Subject Editor)
openscience@royalsociety.org

Associate Editor's comments:

The reviewers are broadly favourable towards your paper, but recommend a number of changes to ensure the manuscript is of a suitable qualitative standard for publication. Please ensure that you incorporate their recommended changes or provide reasonable justification for not doing so in your revision. Good luck and thank you for submitting.

Comments to Author:

Reviewers' Comments to Author:

Reviewer: 1

Comments to the Author(s)

The content is technically correct. However, it is not placed in proper perspective. Some comments are given below:

1. The names of the proposed group of transforms and the resulting analytic signals should be suggestive that they are derived from the sine and cosine transforms.
2. Many signals can have same spectrum, and we know the reason for this fact. Now, one signal is having sixteen different time-frequency distributions. What is the meaning of this revelation?
3. Some discussion on why we need the above transforms will be highly desirable.

Reviewer: 2

Comments to the Author(s)

This reviewer feels that the manuscript is weak in terms of recent literature related to the research work presented in it. The below mentioned references should be included in the manuscript before its acceptance for publication in order to make it very useful for readers and to show the effectiveness of the proposed work with reference to existing methods.

Recently, a new technique for time-frequency analysis which is based on improved eigenvalue decomposition and Wigner-Ville distribution has been proposed for cross-terms free time-frequency representation for multicomponent non-stationary signals. Please see the following paper:

R.R. Sharma and R.B. Pachori, Improved eigenvalue decomposition-based approach for reducing cross-terms in Wigner-Ville distribution, *Circuits, Systems, and Signal Processing*, vol. 37, issue 08, pp. 3330-3350, August 2018.

Authors should discuss this method in the Introduction part of the paper. Also, explain how your proposed method is better than the above mentioned method can be explained.

In the other paper, authors have proposed a new method for time-frequency analysis using eigenvalue decomposition of Hankel matrix for complex signals. Please see below paper for more information.

R.R. Sharma and R.B. Pachori, Eigenvalue decomposition of Hankel matrix-based time-frequency representation of complex signals, *Circuits, Systems, and Signal Processing*, vol. 37, issue 08, pp. 3313-3329, August 2018.

The paper needs to cite this paper and should mention about the extension of the presented approach for time-frequency analysis for complex signals.

Recently, authors have proposed a new approach for time-frequency analysis using Fourier-Bessel series expansion based empirical wavelet transform and normalized Hilbert transform in the following paper:

A. Bhattacharyya, L. Singh, and R.B. Pachori, Fourier-Bessel series expansion based empirical wavelet transform for analysis of non-stationary signals, *Digital Signal Processing*, vol. 78, pp. 185-196, July 2018.

Authors should discuss this approach and developed method in the manuscript.

The other paper in literature develops a new method for time-frequency distribution based on improved eigenvalue decomposition and Hilbert transform.

R.R. Sharma and R.B. Pachori, Time-frequency representation using IEVDHM-HT with application to classification of epileptic EEG signals, *IET Science, Measurement & Technology*, vol. 12, issue 01, pp. 72-82, January 2018.

How the presented method is better than the suggested method in this paper? Please mention it in the paper.

In the literature, time-frequency analysis has been proposed for multivariate signals. Please see below paper.

A. Bhattacharyya and R.B. Pachori, A multivariate approach for patient specific EEG seizure detection using empirical wavelet transform, *IEEE Transactions on Biomedical Engineering*, vol. 64, no. 09, pp. 2003-2015, September 2017.

How the presented method can be extended for multivariate signals. This can be included as a future direction of this work with respect to the above mentioned paper.

In the literature, tunable-Q wavelet transform together with Wigner-Ville distribution has been studied for improving Wigner-Ville distribution based time-frequency analysis in the following paper:

R.B. Pachori and A. Nishad, Cross-terms reduction in Wigner-Ville distribution using tunable-Q wavelet transform, *Signal Processing*, vol. 120, pp. 288-304, March 2016.

How proposed method is different than this method for time-frequency analysis. Please mention in your paper.

Fourier-Bessel series expansion based time-order representation has been proposed for time-frequency analysis for speech signals in the following paper:

P. Jain and R.B. Pachori, Time-order representation based method for epoch detection from speech signals, *Journal of Intelligent Systems*, vol. 21, issue 1, pp. 79-95, February 2012.

How your method is expected to be better than this method? Please explain in the paper.

In another method of time-frequency analysis, Fourier-Bessel series expansion based component separation has been used together with Wigner-Ville distribution for improved time-frequency representation in the below paper.

R.B. Pachori and P. Sircar, A new technique to reduce cross terms in the Wigner distribution, *Digital Signal Processing*, vol. 17, no. 2, pp. 466-474, March 2007.

Please explain your method with respect to the presented method in the above mentioned paper.

Various applications of the developed method have been studied. Same applications have been reported in the below mentioned papers.

Recently, authors have proposed a new method for instantaneous fundamental frequency estimation using eigenvalue decomposition of the Hankel matrix in the following paper:

P. Jain and R.B. Pachori, Event-based method for instantaneous fundamental frequency estimation from voiced speech based on eigenvalue decomposition of Hankel matrix, *IEEE/ACM Transactions on Audio, Speech and Language Processing*, vol. 22, issue 10, pp. 1467-1482, October 2014.

Authors should also discuss the above mentioned paper in the interpretation of the obtained results using the proposed method.

The manuscript has also studied application of the proposed method for baseline wander removal. Recently, a new method based on the eigenvalue decomposition has been proposed for the same application. Please see the below reference.

R.R. Sharma and R.B. Pachori, Baseline wander and power line interference removal from ECG signals using eigenvalue decomposition, Biomedical Signal Processing and Control, vol. 45, pp. 33-49, August 2018.

Please discuss your obtained results for this application with reference to the above mentioned paper.

Reviewer: 3

Comments to the Author(s)

The revised manuscript presents novel Fourier quadrature transforms and analytic signal representations for nonlinear and non-stationary time series analysis. The manuscript appears to be improved considerably by incorporating suggestions made by the reviewers. Although the language of the manuscript is in an acceptable level, it might be useful to get it read by a language expert as there are always some grammatical errors. For example, in page 9, line 4 from the bottom, delete "An" at the beginning of the sentence to start as "Estimation of...". Also, change "a" to "the" in "...is a most important..." to read as "...is the most important...".

Author's Response to Decision Letter for (RSOS-181131.R0)

See Appendix A.

RSOS-181131.R1 (Revision)

Review form: Reviewer 2

Is the manuscript scientifically sound in its present form?

Yes

Are the interpretations and conclusions justified by the results?

Yes

Is the language acceptable?

Yes

Is it clear how to access all supporting data?

Yes

Do you have any ethical concerns with this paper?

No

Have you any concerns about statistical analyses in this paper?

No

Recommendation?

Accept as is

Comments to the Author(s)

The revised paper can be accepted for publication.

Decision letter (RSOS-181131.R1)

01-Nov-2018

Dear Dr Singh,

I am pleased to inform you that your manuscript entitled "Novel Fourier Quadrature Transforms and Analytic Signal Representations for Nonlinear and Non-stationary Time Series Analysis" is now accepted for publication in Royal Society Open Science.

on behalf of Mr Andrew Dunn (Associate Editor) and R. Kerry Rowe (Subject Editor)
openscience@royalsociety.org

Associate Editor Comments to Author (Mr Andrew Dunn):
Associate Editor
Comments to the Author:
(There are no comments.)

Reviewer comments to Author:

Reviewer: 2

Comments to the Author(s)

The revised paper can be accepted for publication.

Appendix A

Response to Referees

Dated: October 27, 2018

Respected Editors,

First of all, please accept our thanks for coordinating the reviews of the manuscript RSOS-181131 entitled "*Novel Fourier Quadrature Transforms and Analytic Signal Representations for Nonlinear and Non-stationary Time Series Analysis*" which we submitted to journal Royal society open science. We submitted this manuscript on July 11, 2018 and received first review comments on October 18, 2018. We really appreciate the time and effort devoted by you and the anonymous reviewers in reviewing and providing useful comments on the manuscript.

We have done our best to address reviewers' comments. We, hereby, submit: (1) a letter of response to referees with detailed replies to each of the reviewers' comments, (2) a revised manuscript with major changes highlighted, and (3) a revised manuscript that we believe to have notably improved after incorporating suggestions of the reviewers.

Please do thank the anonymous reviewers on our behalf for their time and efforts. We have acknowledged the reviewers in the acknowledgement section of this manuscript for their support in helping to improve the quality of the manuscript. We would also like to thank you for your work as an editor.

Thanks, and regards,
Pushpendra Singh

Associate Editor's comments:

The reviewers are broadly favourable towards your paper, but recommend a number of changes to ensure the manuscript is of a suitable qualitative standard for publication. Please ensure that you incorporate their recommended changes or provide reasonable justification for not doing so in your revision. Good luck and thank you for submitting.

Response: Author would like to thank the editor for devoting his time and efforts to coordinate the reviews. We truly appreciate the editor for providing informed decision on time and providing an opportunity to improve the manuscript for possible publication in the reputed journal **Royal society open science**.

Comments to Author:

Reviewers' Comments to Author:

Reviewer: 1

Comments to the Author(s)

The content is technically correct. However, it is not placed in proper perspective. Some comments are given below:

Response: Author would like to thank reviewer for providing comments to improve the manuscript.

1. The names of the proposed group of transforms and the resulting analytic signals should be suggestive that they are derived from the sine and cosine transforms.

Response: The sine and cosine transforms are variants of the Fourier transform for odd and even signal, respectively. Therefore, we designated the proposed transforms in continuous-time domain as "Fourier cosine quadrature transforms (FCQT)" and "Fourier sine quadrature transform (FSQT)" and these FCQT and FSQT are combinedly designated as the Fourier Quadrature Transforms (FQTs). In discrete-

time domain, due to four boundary conditions (even or odd two choices at both left and right end; two choices of symmetric about a data point or the point halfway between two data points) there are sixteen variants of discrete sine and cosine transforms (eight discrete FCQTs and discrete FSQTs). Now, as per suggestion, we have designated eight analytic signals as DCTs based analytic signal representations (DCT-ASRs) and another eight analytic signals as DSTs based analytic signal representations (DST-ASRs), and combinedly designated as discrete Fourier quadrature analytic signal (FQAS) representations. All changes are highlighted in the revised version of the manuscript.

2. Many signals can have same spectrum, and we know the reason for this fact. Now, one signal is having sixteen different time-frequency distributions. What is the meaning of this revelation?

Response: There are two continuous-time sine and cosine transforms and in discrete-time domain, due to four boundary conditions, there are sixteen variants of discrete sine and cosine transforms. Moreover, there is main difference of boundary condition among DCTs and DSTs. Therefore, there is main difference of boundary condition among sixteen time-frequency distributions produced by the proposed method. As the DCTs present energy compaction and decorrelation property, and thus energy should be concentrated more in the low frequency components of time-frequency plane. To obtain the definite revelation of sixteen time-frequency distributions, we need to perform comprehensive study of each time-frequency distribution and this would be future direction of research.

3. Some discussion on why we need the above transforms will be highly desirable.

Response: Many time-frequency-representation approaches such as the empirical mode decomposition (EMD) algorithms and its variants, Fourier decomposition method (FDM), variational mode decomposition (VMD) are primarily based on the Hilbert transform (HT) and Gabor analytic signal (GAS) representation which are well-known and widely-used in numerous applications. Practically, the HT and GAS are implemented using the discrete Fourier transform (DFT) which is efficiently computed using fast Fourier transform (FFT) algorithm. This study proposes sixteen FQTs as alternatives of the HT, sixteen FQAS representations as alternatives of the GAS, and presents detailed simulation results using DCT type-2 based FQT and FQAS which provides better results, e.g., in terms of accurate instantaneous frequency estimation (Figure 2 and 3) and higher output SNR (Figure 5 c) in many but limited applications considered in this work. Moreover, the proposed novel methodology produces better results, provides alternative ways of time-series representation and analysis, advances the existing Fourier decomposition method (FDM) which is useful many applications [1], and thus proposed transforms and corresponding analytic signal representations are needed. Finally, as the DFT and DCTs are widely used tools in numerous different sets of applications, so our future direction of research would be to explore proposed FQTs and corresponding analytic signal representations in various applications. This discussion is presented in the conclusion section of the revised version of the manuscript.

Reviewer: 2

Comments to the Author(s)

This reviewer feels that the manuscript is weak in terms of recent literature related to the research work presented in it. The below mentioned references should be included in the manuscript before its acceptance for publication in order to make it very useful for readers and to show the effectiveness of the proposed work with reference to existing methods.

Response: Author would like to thanks reviewer for providing constructing comments to make the manuscript suitable for publishing.

Recently, a new technique for time-frequency analysis which is based on improved eigenvalue decomposition and Wigner-Ville distribution has been proposed for cross-terms free time-frequency representation for multicomponent non-stationary signals. Please see the following paper:

R.R. Sharma and R.B. Pachori, Improved eigenvalue decomposition-based approach for reducing cross-terms in Wigner-Ville distribution, *Circuits, Systems, and Signal Processing*, vol. 37, issue 08, pp. 3330-3350, August 2018.

Authors should discuss this method in the Introduction part of the paper. Also, explain how your proposed method is better than the above mentioned method can be explained.

Response: As per suggestion, the above paper has been discussed in the introduction part of the paper. The above paper is based on the Wigner-Ville distribution which suffers from cross terms interference, however they are reduced (but not eliminated) in the mentioned study. On the other hand, the Fourier decomposition method (FDM) is based on the Fourier theory and zero-phase filtering [1,2]. The FDM can decompose real as well as complex signals (which can be multichannel or multivariate) into a set of desired number of Fourier intrinsic band functions (FIBFs) with desired cutoff frequencies.

In the other paper, authors have proposed a new method for time-frequency analysis using eigenvalue decomposition of Hankel matrix for complex signals. Please see below paper for more information.

R.R. Sharma and R.B. Pachori, Eigenvalue decomposition of Hankel matrix-based time-frequency representation of complex signals, *Circuits, Systems, and Signal Processing*, vol. 37, issue 08, pp. 3313-3329, August 2018.

The paper needs to cite this paper and should mention about the extension of the presented approach for time-frequency analysis for complex signals.

Response: As per suggestion, the above paper has been discussed in the introduction part of the paper. As mentioned above, the FDM can produce time-frequency distribution (TFD) of complex signals, as analytic FIBFs are obtained in FDM and for a complex signal sum of FIBFs is the original complex signal.

Recently, authors have proposed a new approach for time-frequency analysis using Fourier-Bessel series expansion based empirical wavelet transform and normalized Hilbert transform in the following paper:

A. Bhattacharyya, L. Singh, and R.B. Pachori, Fourier-Bessel series expansion based empirical wavelet transform for analysis of non-stationary signals, *Digital Signal Processing*, vol. 78, pp. 185-196, July 2018.

Authors should discuss this approach and developed method in the manuscript.

Response: As per suggestion, the above paper has been discussed in the introduction part of the paper

The other paper in literature develops a new method for time-frequency distribution based on improved eigenvalue decomposition and Hilbert transform.

R.R. Sharma and R.B. Pachori, Time-frequency representation using IEVDHM-HT with application to classification of epileptic EEG signals, *IET Science, Measurement & Technology*, vol. 12, issue 01, pp. 72-82, January 2018.

How the presented method is better than the suggested method in this paper? Please mention it in the paper.

Response: As per suggestion, the above paper has been discussed in the introduction part of the paper. The FDM is computationally efficient due to FFT implementation, produces desired number of FIBFs with required cutoff frequencies, however, these features are difficult to achieve using IEVDHM-HT.

In the literature, time-frequency analysis has been proposed for multivariate signals. Please see below paper.

A. Bhattacharyya and R.B. Pachori, A multivariate approach for patient specific EEG seizure detection using empirical wavelet transform, IEEE Transactions on Biomedical Engineering, vol. 64, no. 09, pp. 2003-2015, September 2017.

How the presented method can be extended for multivariate signals. This can be included as a future direction of this work with respect to the above mentioned paper.

Response: As per suggestion, the above paper has been discussed in the introduction part of the paper as an example of time-frequency application. Thanks for suggestion, as already mentioned, presented method i.e. FDM is valid for multichannel or multivariate signals, and future direction of research would be to explore the application of FDM for patient specific EEG seizure detection.

In the literature, tunable-Q wavelet transform together with Wigner-Ville distribution has been studied for improving Wigner-Ville distribution based time-frequency analysis in the following paper:

R.B. Pachori and A. Nishad, Cross-terms reduction in Wigner-Ville distribution using tunable-Q wavelet transform, Signal Processing, vol. 120, pp. 288-304, March 2016.

How proposed method is different than this method for time-frequency analysis. Please mention in your paper.

Response: As per suggestion, the above paper has been discussed in the introduction part of the paper as an example of time-frequency method. Above paper is based on tunable-Q wavelet transform with Wigner-Ville distribution for time-frequency analysis of signal, however proposed method is based on the Fourier theory (i.e. DCTs, DSTs and DFT).

Fourier-Bessel series expansion based time-order representation has been proposed for time-frequency analysis for speech signals in the following paper:

P. Jain and R.B. Pachori, Time-order representation based method for epoch detection from speech signals, Journal of Intelligent Systems, vol. 21, issue 1, pp. 79-95, February 2012.

How your method is expected to be better than this method? Please explain in the paper.

Response: As per suggestion, the above paper has been discussed in the introduction part of the paper as an example of time-frequency application for epoch detection from speech signals. The future direction of research would be to explore the application of FDM for epoch detection from speech signals.

In another method of time-frequency analysis, Fourier-Bessel series expansion based component separation has been used together with Wigner-Ville distribution for improved time-frequency representation in the below paper.

R.B. Pachori and P. Sircar, A new technique to reduce cross terms in the Wigner distribution, Digital Signal Processing, vol. 17, no. 2, pp. 466-474, March 2007.

Please explain your method with respect to the presented method in the above mentioned paper.

Response: As per suggestion, the above paper has been discussed in the introduction part of the paper as an example of time-frequency method which reduces cross terms in the Wigner distribution. The FDM is a different method for time-frequency representation than the Wigner distribution approach.

Various applications of the developed method have been studied. Same applications have been reported in the below mentioned papers.

Recently, authors have proposed a new method for instantaneous fundamental frequency estimation using eigenvalue decomposition of the Hankel matrix in the following paper:

P. Jain and R.B. Pachori, Event-based method for instantaneous fundamental frequency estimation from voiced speech based on eigenvalue decomposition of Hankel matrix, IEEE/ACM Transactions on Audio, Speech and Language Processing, vol. 22, issue 10, pp. 1467-1482, October 2014.

Authors should also discuss the above mentioned paper in the interpretation of the obtained results using the proposed method.

Response: As per suggestion, the above paper has been discussed in the introduction part of the paper as an example of time-frequency application for instantaneous fundamental frequency estimation from voiced speech. Instantaneous fundamental frequency estimation results obtained from the FDM are compared with suggested method.

The manuscript has also studied application of the proposed method for baseline wander removal. Recently, a new method based on the eigenvalue decomposition has been proposed for the same application. Please see the below reference.

R.R. Sharma and R.B. Pachori, Baseline wander and power line interference removal from ECG signals using eigenvalue decomposition, Biomedical Signal Processing and Control, vol. 45, pp. 33-49, August 2018.

Please discuss your obtained results for this application with reference to the above mentioned paper.

Response: As per suggestion, the above paper has been discussed in the introduction part of the paper as an example of time-frequency application for Baseline wander and power line interference removal from ECG signals, and we discussed the results obtained from the proposed method with respect to this suggested paper.

Reviewer: 3

Comments to the Author(s)

The revised manuscript presents novel Fourier quadrature transforms and analytic signal representations for nonlinear and non-stationary time series analysis. The manuscript appears to be improved considerably by incorporating suggestions made by the reviewers. Although the language of the manuscript is in an acceptable level, it might be useful to get it read by a language expert as there are always some grammatical errors. For example, in page 9, line 4 from the bottom, delete "An" at the beginning of the sentence to start as "Estimation of...". Also, change "a" to "the" in "...is a most important..." to read as "...is the most important...".

Response: Author would like to thank reviewer for providing comments on the proposed methods and also for suggesting to improve the language of the paper. As suggested, we have got the revised manuscript read by a language expert and filtered out such mistakes.

Journal Name: Royal Society Open Science

Journal Code: RSOS

Online ISSN: 2054-5703

Journal Admin Email: openscience@royalsociety.org

Journal Editor: Emilie Aime

Journal Editor Email: emilie.aime@royalsociety.org

MS Reference Number: RSOS-181131

Article Status: SUBMITTED

MS Dryad ID: RSOS-181131

MS Title: Novel Fourier Quadrature Transforms and Analytic Signal Representations for Nonlinear and Non-stationary Time Series Analysis

MS Authors: Singh, Pushpendra

Contact Author: Pushpendra Singh

Contact Author Email: spushp@gmail.com

References:

[1] P. Singh, S.D. Joshi, R.K. Patney, K. Saha, The Fourier decomposition method for nonlinear and non-stationary time series analysis. Proc. R. Soc. A 20160871 (2017).

<http://dx.doi.org/10.1098/rspa.2016.0871>.

[2] P. Singh, Breaking the Limits: Redefining the Instantaneous Frequency, Circuits Syst Signal Process, 37 (8), 3515--3536 (2018). <https://doi.org/10.1007/s00034-017-0719-y>.